# Predicting short-term mortality in severe cirrhosis: An interpretable machine learning model integrating routine clinical indicators

Shun Zhang[1,2◎], Rui Liu[2◎], Zhengjie Li[1,2◎], Tao Pan[2*], Xudong Wen[2*]

1 Department of Gastroenterology, Chengdu University of Traditional Chinese Medicine Affiliated Hospital of Integrated Traditional Chinese and Western Medicine, Chengdu, China, 2 Department of Gastroenterology, Chengdu Integrated TCM & Western Medicine Hospital, Chengdu University of Traditional Chinese Medicine, Chengdu, China,

◎ These authors contributed equally to this work.
* pant414@163.com (TP); xudongwen@cdutcm.edu.cn (XW)

## Abstract

### Background

The critical need for precise risk stratification in severe liver cirrhosis is underscored by its substantial 30-day mortality rates, demanding reliable tools to guide clinical interventions.

### Objective

To establish a machine learning-driven prognostic model for short-term mortality prediction in decompensated cirrhosis through comprehensive analysis of critical care data.

### Methods

This retrospective cohort study analyzed 1,044 carefully curated cases from the MIMIC-IV database, randomly divided into training (n = 740) and validation (n = 304) sets. We developed a machine learning model incorporating multidimensional clinical parameters, with rigorous evaluation and internal validation. Short-term survival was analyzed via bootstrap-validated Cox proportional hazards regression. Prognostic heterogeneity across international normalized ratio (INR)-based strata was examined.

### Results

The final prediction model incorporated eight significant predictors: age (OR 1.051, 95% CI 1.033–1.070), INR (OR 1.423, 95%CI 1.231–1.644), creatinine (OR 1.171, 95%CI 1.071–1.208), platelets (OR 0.995, 95%CI 0.993–0.997), white blood cell (OR 1.116, 95%CI 1.078–1.155), total bilirubin (OR 1.027, 95%CI 1.002–1.052), peptic

**Data availability statement:** The data underlying the results presented in the study are available from (https://physionet.org/content/mimiciv/).

**Funding:** This study was supported by National Natural Science Foundation of China (grant number 82474299).

**Competing interests:** The authors have declared that no competing interests exist.

ulcer (OR 0.336, 95%CI 0.134–0.845), and Aspartate Aminotransferase/Alanine Aminotransferase (AST/ALT) (OR 1.508, 95%CI 1.294–1.757). The model demonstrated excellent discrimination with an AUC of 0.846 in the training cohort. Cox regression analysis confirmed these findings and identified additional associations with aspartate aminotransferase and red blood cell levels. Furthermore, the indicators within the model provide accurate predictions for the clinical outcomes of patients suffering from severe cirrhosis. Subgroup analysis revealed significant mortality variations across different INR ranges ($P < 0.001$).

## Conclusions

Our prediction model identifies high-risk cirrhotic patients and highlights critical prognostic factors, offering clinicians a valuable tool for risk stratification and timely intervention. The strong correlation between laboratory markers, complications, and outcomes underscores the importance of close monitoring in this population. However, our model is an initial step, effective within the ICU but requiring external, multi-center studies to broaden its clinical applicability, which is a clear priority for our future work.

---

## Introduction

Liver cirrhosis constitutes a major global health challenge, currently ranking as the 14th leading cause of mortality worldwide. Recent epidemiological data reveal an alarming annual death toll of 1.2 million individuals, with incidence rates projected to escalate through 2039 [1,2]. In liver cirrhosis, the high mortality rate is primarily attributable to episodes of acute or chronic liver failure (ACLF) that mark the decompensated stage [3] advanced stages demonstrate disproportionately high mortality rates attributable to complex pathophysiological mechanisms including immune dysregulation, systemic inflammatory responses, and heightened susceptibility to infections [4,5]. Contemporary cohort studies document particularly concerning short-term outcomes: 3-month mortality rates reach 54–58% in decompensated cirrhosis populations [6], while infected cirrhotic patients demonstrate a 63% 1-year mortality rate with a 3.75-fold increased risk compared to non-infected counterparts [7]. The most severe manifestation, ACLF, carries a striking 28-day mortality rate of 66.97% [8]. This escalating mortality burden generates substantial multidimensional impacts, encompassing psychological distress for patients and families, economic strain on healthcare systems, and significant public health challenges. These converging factors underscore the critical imperative for developing precise short-term prognostic models to optimize clinical decision-making and resource allocation.

Despite the availability of multiple prognostic scoring systems, significant methodological limitations hinder their clinical utility in predicting short-term outcomes. The widely adopted MELD/MELD-Na scores, while validated for end-stage survival prediction, demonstrate limited sensitivity in detecting early decompensation events [9,10]. The Chronic Liver Failure – Organ Failure(CLIF-OF) score is used to predict

outcomes in patients with severe cirrhosis, assessing six major organ systems—liver, kidneys, coagulation, brain, respiration, and circulation—to provide a comprehensive evaluation of systemic condition [11]. Based on this framework, the CLIF Consortium Acute-on-Chronic Liver Failure Score (CLIF-C ACLF) further incorporates age and white blood cell count, reflecting physiological reserve and infection status. Studies have shown that the CLIF-C ACLF score outperforms CTP, MELD, and MELD-Na in predicting both 28-day (AUROC 0.799±0.078; 95% CI 0.637–0.891) and 90-day mortality (AUROC 0.828±0.063; 95% CI 0.705–0.952) in patients with ACLF [12]. Although these models perform well from different perspectives, a previous study indicated that their area under the ROC curve did not exceed 0.8, indicating that further refinement and multi-faceted optimization could enhance prognostic accuracy [13].

In addition to established prognostic scoring systems, a range of individual and composite indicators have been identified as predictors of mortality in severe cirrhosis. Hematologic parameters such as thrombocytopenia and leukocytosis—reflected in platelet and white blood cell (WBC) counts—have been significantly correlated with in-hospital mortality [14]. In decompensated cirrhosis requiring transplantation, the coexisting risks of hemorrhage and thrombosis highlight the critical need to monitor coagulation metrics, including international normalized ratio (INR) and platelet levels. Interventions aimed at rebalancing the procoagulant, anticoagulant, and fibrinolytic pathways may improve outcomes [15]. Liver-specific biomarkers also offer prognostic insight. A multicenter study of patients with sepsis and acute liver failure identified hypoalbuminemia (OR = 0.856, 95% CI: 0.736–0.996) and an elevated Aspartate Aminotransferase/Alanine Aminotransferase(AST/ALT) ratio (OR = 2.018, 95% CI: 1.137–3.580) as independent predictors of in-hospital mortality, with an AST/ALT ratio ≥1.26 exhibiting 90.2% specificity for fatal outcomes [16]. Furthermore, elevated bilirubin levels and altered creatinine—reflecting impaired hepatic and renal/metabolic function, respectively—are consistently associated with poorer prognosis [17]. Baseline demographic and clinical characteristics also contribute to risk stratification. Advanced age (65–79 years) is linked to higher mortality and more frequent complications in end-stage liver disease [18]. Sex and race further modulate disease progression and treatment suitability [19], while declining weight and BMI correlate with worsening liver function and increased mortality, underscoring their utility as prognostic indicators [20]. The clinical course of cirrhosis is frequently complicated by multisystem organ dysfunction, a process largely driven by portal hypertension and culminating in life-threatening complications such as variceal hemorrhage, ascites, and hepatic encephalopathy. [21–23]. It also promotes vascular dysfunction and fluid imbalance, which can lead to refractory hypoxemia and dyspnea. Concurrent syndromes such as hepatopulmonary syndrome (HPS) and portopulmonary hypertension (PoPH) exacerbate cardiopulmonary decline, significantly elevating mortality [24]. Notably, peptic ulcer bleeding exerts an adverse impact on in-hospital outcomes comparable to that of variceal bleeding, further emphasizing its prognostic relevance [25]. Despite these advancements, there remains a pronounced gap in validated tools designed specifically for predicting short-term mortality in critically ill cirrhotic patients. Existing models, though informative, would benefit from the integration of broader clinical variables and larger datasets to enhance accuracy and generalizability.

The aforementioned studies indicate that the factors associated with mortality in liver cirrhosis are diverse and multifaceted, which can primarily be summarized as follows: First, significant differences in cirrhosis-related mortality are observed across demographic variables such as age, gender, and ethnicity [26]. Second, the impairment of the liver's synthetic, metabolic, and detoxification functions triggers a cascade of pathophysiological disturbances. These include aberrant protein synthesis, altered coagulation factor production, elevated liver enzymes, dysregulated bilirubin metabolism, fluid and electrolyte imbalances, and compromised immune function. Consequently, these systemic abnormalities manifest as severe complications like hepatorenal syndrome, hepatic encephalopathy, portal hypertension, and bleeding from gastroesophageal varices, which can progress to multiple organ failure, thereby becoming life-threatening [21]. Additionally, although comorbidities are not direct causes or consequences of cirrhosis, evidence suggests that they significantly affect patient prognosis and increase mortality risk [27]. While several scoring models are available to evaluate disease severity in patients with end-stage acute or chronic liver failure, each assesses the condition from different perspectives. This study aims to integrate demographic information, laboratory parameters, comorbidities, and relevant physiological

and pathological scores to develop a comprehensive model for predicting short-term mortality in cirrhotic patients during acute or chronic liver failure. Through comparative validation across heterogeneous patient cohorts, we aim to quantitatively establish the model's superior predictive accuracy for 30- and 90-day mortality endpoints relative to conventional scoring systems. As a translational objective, we will develop and deploy an open-access clinical decision support interface with real-time risk stratification capabilities, specifically engineered to facilitate early targeted interventions in high-risk subpopulations.

## Methods

### Data source

We conducted a retrospective cohort study using data from the Medical Information Mart for Intensive Care (MIMIC-IV, version 2.0), a large, freely-available critical care database developed through a collaboration between the Massachusetts Institute of Technology (MIT), Beth Israel Deaconess Medical Center, and Philips Healthcare. After completing the required Collaborative Institutional Training Initiative certification and passing the Protecting Human Research Participants exam, we were granted access to this de-identified database, which waived the requirement for informed consent. We obtained the research data from the MIMIC – IV database on March 8, 2025, and all authors declare that the data we obtained cannot identify the personal characteristics of patients.

### Study design and participant selection

Our retrospective analysis utilized hospitalization records from 10,620 patients with cirrhosis diagnoses in the MIMIC-IV critical care database. We implemented rigorous exclusion criteria as detailed: (1) exclusion of non-ICU admissions (n = 334), (2) elimination of repeat hospitalizations through retention of only the most recent admission per patient (n = 7,140 excluded), and (3) removal of cases with missing critical prognostic variables (n = 2,770 excluded), (4) Exclude patients with liver cirrhosis who have malignant tumors. The resultant final analytic cohort consisted of 1,044 unique patients, who underwent stratified randomization via computer-generated allocation sequences to ensure cohort comparability. This process yielded a derivation cohort (n = 740) for model development and an independent validation cohort (n = 304) for performance evaluation, maintaining a 7:3 ratio consistent with machine learning validation standards (Fig 1).

### Data collection and processing

Data extraction and harmonization were conducted using Navicat Premium 16 (PremiumSoft CyberTech Ltd), with comprehensive variable mapping from the MIMIC-IV database. diseaseOur structured data collection framework encompassed five core domains: (1) Demographic characteristics including age (years) and admission weight (kg) at ICU entry; (2) Laboratory parameters assessed during ICU stay, encompassing alkaline phosphatase, alanine aminotransferase, aspartate aminotransferase, international normalized ratio, creatinine, hemoglobin, total bilirubin, potassium, blood urea nitrogen, white blood cell count, red blood cell count, platelet count, albumin (minimum value), and sodium (minimum value); (3) Survival-related information covering ICU admission time, discharge time, time of death, alcohol-related liver disease status, use of anticoagulant therapy, and frequency of prior cirrhosis-related hospitalizations; (4) Disease severity scores including the Model for End-Stage Liver Disease (MELD), CLIF-OF, and CLIF-C ACLF; (5) Comorbid conditions based on the Elixhauser framework, including myocardial infarction, congestive heart failure, peripheral vascular disease, cerebrovascular disease, dementia, chronic pulmonary disease, peptic ulcer disease, liver disease, diabetes, paralysis, renal disease, malignancy, and metastatic solid tumor. Variables exceeding 20% missingness thresholds were systematically excluded per predefined protocol. All statistical analyses were performed using SPSS (version 29.0; IBM Corp, Armonk, NY), Stata 17 (StataCorp LP, College Station, TX), and R 4.3.1 (R Foundation for Statistical Computing, Vienna, Austria), with X-tile 3.6.2 (Yale University) facilitating optimal cutoff determination through adaptive cohort partitioning.

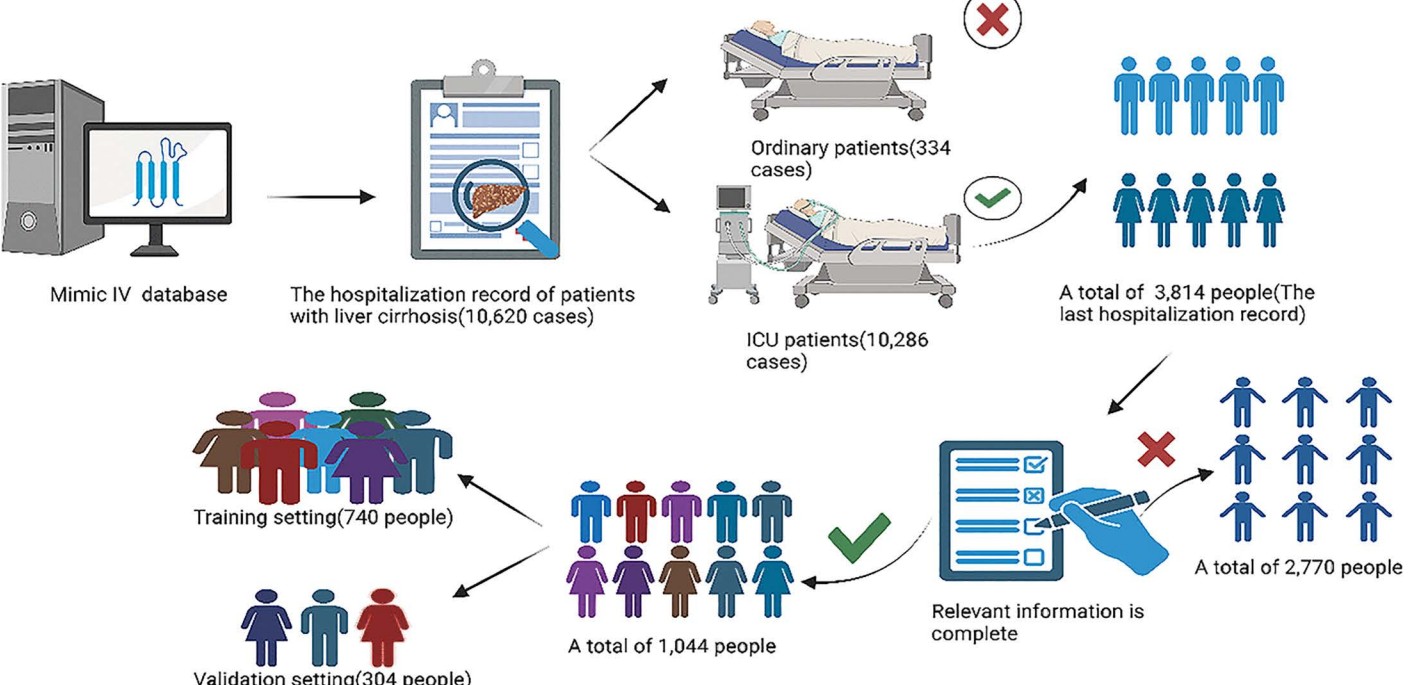

**Fig 1. Study cohort selection and data curation workflow.**

## Primary endpoint

30-day all-cause mortality post-ICU admission.

## Statistical analysis

Continuous variables underwent normality assessment via Kolmogorov-Smirnov tests. Normally distributed variables were expressed as mean±standard deviation and analyzed using independent samples t-tests with homogeneity of variance verification; non-normally distributed variables were reported as median (interquartile range) and compared with Mann-Whitney U tests. In the final variable selection process, candidate variables were initially screened based on clinical relevance. Subsequently, univariate and multivariate logistic regression analyses were performed. Variables retained after these steps were further assessed using LASSO regression, a data-driven technique, which ultimately identified eight variables for inclusion in the final model (S1 Fig). A logistic regression model predicting short-term mortality in cirrhotic patients was constructed incorporating age, INR, creatinine, platelet count, white blood cell count, total bilirubin, peptic ulcer disease, and AST/ALT. Model performance was evaluated through calibration and discrimination (receiver operating characteristic curve) analyses in the derivation cohort, with subsequent external validation in the independent cohort. A nomogram was developed for visual representation. Decision curve analysis (DCA) quantified clinical utility, while comparative discrimination analyses against MELD、MELD-Na、CLIF-OF、and CLIF-C ACLF scores demonstrated superior sensitivity and specificity. Based on a post-hoc power analysis using the observed AUC of 0.851, our study achieved a statistical power of 1.0, indicating that the sample size was fully adequate to detect the model's discriminative performance with high confidence. Internal validation was performed to ensure model robustness. For short-term survival analysis, Cox proportional hazards regression with bootstrap validation (1,000 resamples) was employed. Stratified analyses based on INR thresholds were conducted using Kaplan-Meier methodology to examine mortality incidence across subgroups.

## Results

### Baseline characteristics

The study included 1,044 patients divided into derivation (n = 740) and validation cohorts (n = 304). In the comparison of baseline characteristics between the training and validation cohorts, all variables except for the INR and the presence of renal disease showed no significant differences ($P > 0.05$). Although INR and renal disease were imbalanced between the two groups ($P < 0.05$), the randomized allocation procedure minimized systematic bias in their distribution. Subsequent analyses—including calibration curves (C-index), Receiver operating characteristic (ROC) analysis, and adjustment for anticoagulant use—demonstrated that INR did not adversely affect model performance in either cohort. Furthermore, renal disease was not incorporated into the final predictive model. Taken together, these findings support the appropriateness of the data partitioning and the validity of the validation cohort for model assessment (Table 1).

In the derivation cohort, 204 deaths (27.6%) occurred within 30 days. Compared with survivors, deceased patients exhibited significantly higher age, weight, liver enzymes (ALT, ALP, AST), INR, AST/ALT, creatinine, WBC, blood urea nitrogen (BUN), and total bilirubin, alongside lower hemoglobin, albumin, red blood cell count (RBC), sodium, and platelet levels (all $P < 0.05$). Comorbidities including myocardial infarction, congestive heart failure, cerebrovascular disease, chronic pulmonary disease, renal disease, malignancy, liver disease, and metastatic solid tumors were more prevalent in the death group ($P < 0.05$) (Table 2).

### Model development

Univariate logistic regression identified 13 mortality predictors ($P < 0.05$): age, albumin, sodium, RBC, ALP, AST, AST/ALT, INR, creatinine, platelets, WBC, BUN, total bilirubin (clinical variables), and cerebrovascular disease, peptic ulcer disease, renal disease, and metastatic solid tumors (comorbidities). Additionally, univariate regression was performed for the number of previous hospitalizations due to cirrhosis, which demonstrated a protective association (OR = 0.681, 95% CI: 0.592–0.784, $P < 0.001$). However, as this variable reflects historical healthcare exposure and frequency of medical contact rather than a modifiable physiological state, its effect is highly susceptible to confounding biases. Therefore, we decided not to include it in the multivariable analysis.

Multivariable logistic regression identified eight independent predictors of 30-day mortality in severe cirrhosis (Table 3; visualized via nomogram in Fig 2A): age (adjusted odds ratio [aOR] 1.051, 95% CI 1.033–1.070 per year), INR; aOR 1.423, 95% CI 1.231–1.644 per unit), creatinine (aOR 1.171, 95% CI 1.071–1.280 mg/dL), platelets (aOR 0.995, 95% CI 0.993–0.997 × 10³/μL), WBC (aOR 1.116, 95% CI 1.078–1.155 × 10³/μL), total bilirubin (aOR 1.027, 95% CI 1.002–1.052 mg/dL), AST/ALT (aOR 1.508, 95% CI 1.294–1.757), and peptic ulcer disease (aOR 0.336, 95% CI 0.134–0.845),. Key clinical interpretations revealed: (1) 5.1% increase in short-term mortality risk per additional year of age ($P < 0.001$); (2) 42.3% increase per unit rise in INR ($P < 0.001$); (3) 50.8% increase per unit increase in AST/ALT ratio ($P < 0.001$). Elevated creatinine, elevated white blood cell count, increased total bilirubin, and decreased platelet count were also significantly associated with increased short-term mortality (all $P < 0.05$). Notably, the presence of peptic ulcer disease was associated with a substantially reduced risk of short-term mortality (aOR 0.451, $P < 0.001$; potential confounding and mechanistic explanations discussed in Limitations). The nomogram achieved clinically actionable stratification (Fig 2B), with low-risk (<6 points) and high-risk (>18 points) groups demonstrating <0.1% and 99% predicted mortality, respectively.

### Model performance evaluation

**Model calibration.** The nomogram demonstrated high calibration accuracy for 30-day mortality prediction in severe cirrhosis across both derivation and validation cohorts. Quantitative assessments revealed: (1) observed-to-expected mortality ratio of 1.00, indicating near-perfect agreement between predicted and actual outcomes; (2) calibration-in-the-large intercept of 0, confirming absence of systematic over- or under-prediction bias. Visual calibration plots further

**Table 1. Characteristics at baseline of patients in the study.**

| Variable | Training Setting (n = 740) | Validation Setting (n = 304) | P Value |
|---|---|---|---|
| Age (years) | 60.24 ± 12.22 | 59.54 ± 12.16 | 0.403 |
| Weight (kg) | 84.99 ± 21.48 | 85.25 ± 20.32 | 0.856 |
| Hemoglobin (max, g/dl) | 10.21 ± 1.96 | 10.04 ± 1.88 | 0.192 |
| Albumin (min, g/dl) | 2.71 ± 0.56 | 2.69 ± 0.55 | 0.650 |
| RBC (max, 10^6/ul) | 3.26 ± 0.67 | 3.21 ± 0.68 | 0.299 |
| Sodium (min, mmol/l) | 132.28 ± 6.12 | 132.08 ± 5.57 | 0.614 |
| ALT (max, u/l) | 58.00 (30.00-153.00) | 61.00 (30.25-121.25) | 0.480 |
| ALP (max, u/l) | 110.00 (74.00-152.47) | 115.00 (77.00-155.00) | 0.278 |
| AST (max, u/l) | 112.50 (57.00-289.50) | 104.5 (51.00-219.25) | 0.331 |
| INR (max) | 1.80 (1.40-2.70) | 2.00 (1.50-3.18) | 0.038 |
| Creatinine (max, mg/dl) | 1.60 (0.90-3.00) | 1.60 (1.00-3.00) | 0.625 |
| Platelet (max, 10^9/l) | 150.00 (102.00-240.75) | 157.00 (101.25-234.50) | 0.758 |
| WBC (max, 10^9/l) | 9.30 (7.73-9.80) | 9.40 (7.60-9.80) | 0.507 |
| BUN (max, mg/dl) | 37.00 (22.00-65.00) | 39.50 (21.00-69.50) | 0.708 |
| Bilirubin total (max, mg/dl) | 3.60 (1.50-8.40) | 3.80 (1.80-9.08) | 0.143 |
| ALT/AST | 1.97(1.40-2.74) | 1.97(1.37-2.78) | 0.848 |
| Number of hospitalizations | 1.00(1.00-3.00) | 1.00(1.00-3.00) | 0.899 |
| **Comorbidity** | | | |
| Myocardial infarct | | | |
| Yes | 42 (5.7%) | 20 (6.6%) | 0.575 |
| No | 698 (94.3%) | 284 (93.4%) | |
| Congestive heart failure | | | |
| Yes | 146 (19.7%) | 49 (16.1%) | 0.174 |
| No | 594 (80.3%) | 255 (83.9%) | |
| Peripheral vascular disease | | | |
| Yes | 50 (6.8%) | 30 (9.9%) | 0.086 |
| No | 690 (93.2%) | 274 (90.1%) | |
| Cerebrovascular disease | | | |
| Yes | 54 (7.3%) | 25 (8.2%) | 0.607 |
| No | 686 (92.7%) | 279 (91.8%) | |
| Chronic pulmonary disease | | | |
| Yes | 181 (24.5%) | 75 (24.7%) | 0.942 |
| No | 559 (75.5%) | 229 (75.3%) | |
| Peptic ulcer disease | | | |
| Yes | 56 (7.6%) | 15 (4.9%) | 0.125 |
| No | 684 (92.4%) | 289 (95.1%) | |
| Renal disease | | | |
| Yes | 172 (23.2%) | 53 (17.4%) | 0.038 |
| No | 568 (76.8%) | 251 (82.6%) | |
| Severe liver disease | | | |
| Yes | 430 (58.1%) | 181 (59.5%) | 0.670 |
| No | 310 (41.9%) | 123 (40.5%) | |

[a]P value: probability value; [b]RBC: red blood cell; [c]ALT: alanine aminotransferase; [d]ALP: alkaline Phosphatase; [e]AST: aspartate aminotransferase; [f]INR: international normalized ratio; [g]WBC: white blood cell; [h]BUN: blood urea nitrogen.

**Table 2. Baseline analysis of the training cohort.**

| Variable | Total Patients (n=740) | Survival (n=536) | Dead (n=204) | P value |
|---|---|---|---|---|
| Age (years) | 60.24±12.22 | 59.39±12.3 | 62.46±11.76 | 0.002 |
| Weight (kg) | 84.99±21.48 | 84.30±21.43 | 86.80±21.57 | 0.157 |
| Hemoglobin (max, g/dl) | 10.21±1.96 | 10.29±1.90 | 10.01±2.12 | 0.075 |
| Albumin (min, g/dl) | 2.71±0.56 | 2.75±0.55 | 2.60±0.56 | 0.001 |
| RBC (max, 10^6/ul) | 3.26±0.67 | 3.30±0.65 | 3.14±0.72 | 0.003 |
| Sodium (min, mmol/l) | 132.28±6.12 | 132.62±5.48 | 131.4±7.51 | 0.035 |
| ALT (max, u/l) | 58.00 (30.00-153.00) | 56.50 (30.00-143.30) | 66.50 (34.00-189.61) | 0.130 |
| ALP (max, u/l) | 110.00 (74.00-152.47) | 109.00 (71.00-142.75) | 121.00 (79.00-168.50) | 0.015 |
| AST (max, u/l) | 112.50 (57.00-289.50) | 100.00 (51.25-238.00) | 169.50 (74.00-470.03) | 0.000 |
| INR (max) | 1.80 (1.40-2.70) | 1.70 (1.30-2.30) | 2.60 (1.90-3.70) | 0.000 |
| Creatinine (max, mg/dl) | 1.60 (0.90-3.00) | 1.20 (0.83-2.30) | 2.65 (1.73-4.58) | 0.000 |
| Platelet (max, 10^9/l) | 150.00 (102.00-240.75) | 158.00 (107.25-243.00) | 134.00 (91.00-211.00) | 0.004 |
| WBC (max, 10^9/l) | 9.30 (7.73-9.80) | 9.00 (7.30-9.70) | 9.60 (8.60-18.15) | 0.000 |
| Bun (max, mg/dl) | 37.00 (22.00-65.00) | 30.50(19.00-56.00) | 54.50 (35.25-87.00) | 0.000 |
| Bilirubin total (max, mg/dl) | 3.60 (1.50-8.40) | 3.30 (1.30-7.20) | 6.55 (3.20-9.78) | 0.000 |
| AST/ALT | 1.97(1.40-2.74) | 1.78(1.29-2.48) | 2.63(1.88-3.60) | 0.000 |
| Number of hospitalizations | 1.00(1.00-3.00) | 2.00(1.00-3.00) | 1.00(1.00-2.00) | 0.000 |
| **Co-morbidity** | | | | |
| Myocardial infarct | | | | |
| Yes | 42 (5.7%) | 26 (4.9%) | 16 (7.8%) | 0.116 |
| No | 698 (94.3%) | 510 (95.1%) | 188 (92.2%) | |
| Congestive heart failure | | | | |
| Yes | 146 (19.7%) | 102 (19%) | 44 (21.6%) | 0.438 |
| No | 594 (80.3%) | 434 (81%) | 160 (78.4%) | |
| Peripheral vascular disease | | | | |
| Yes | 50 (6.8%) | 37 (6.9%) | 13 (6.4%) | 0.797 |
| No | 690 (93.2%) | 499 (93.1%) | 191 (93.6%) | |
| Cerebrovascular disease | | | | |
| Yes | 54 (7.3%) | 32 (6%) | 22 (10.8%) | 0.024 |
| No | 686 (92.7%) | 504 (94%) | 182 (89.2%) | |
| Chronic pulmonary disease | | | | |
| Yes | 181 (24.5%) | 137 (25.6%) | 44 (21.6%) | 0.259 |
| No | 559 (75.5%) | 399 (74.4%) | 160 (78.4%) | |
| Peptic ulcer disease | | | | |
| Yes | 56 (7.6%) | 48 (9%) | 8 (3.9%) | 0.021 |
| No | 684 (92.4%) | 488 (91%) | 196 (96.1%) | |
| Renal disease | | | | |
| Yes | 172 (23.2%) | 109 (20.3%) | 63 (30.9%) | 0.002 |
| No | 568 (76.8%) | 427 (79.7%) | 141 (69.1%) | |
| Severe liver disease | | | | |
| Yes | 430 (58.1%) | 304 (56.7%) | 126 (61.8%) | 0.214 |
| No | 310 (41.9%) | 232 (43.3%) | 78 (38.2%) | |

[a]P value: probability value; [b]RBC: red blood cell; [c]ALT: alanine aminotransferase; [d]ALP: alkaline Phosphatase; [e]AST: aspartate aminotransferase; [f]INR: international normalized ratio; [g]WBC: white blood cell; [h]BUN: blood urea nitrogen.

**Table 3. Univariate and multivariate Logistic regression analyses.**

| Variables | Univariate logistic model | | | Multivariable logistic model | | |
|---|---|---|---|---|---|---|
| | OR | 95%CI | *P* value | OR | 95%CI | *P* Value |
| Age | 1.021 | 1.007-1.035 | 0.002 | 1.051 | 1.033-1.070 | 0.000 |
| ALP | 1.002 | 1.001-1.004 | 0.004 | | | |
| AST | 1.000 | 1.000-1.000 | 0.000 | | | |
| INR | 1.653 | 1.452-1.883 | 0.000 | 1.423 | 1.231-1.644 | 0.000 |
| Creatinine | 1.260 | 1.171-1.356 | 0.000 | 1.171 | 1.071-1.280 | 0.001 |
| Platelet | 0.998 | 0.997-1.000 | 0.019 | 0.995 | 0.993-997 | 0.000 |
| RBC | 0.686 | 0.532-0.885 | 0.004 | | | |
| WBC | 1.110 | 1.078-1.143 | 0.000 | 1.116 | 1.078-1.155 | 0.000 |
| Albumin | 0.610 | 0.453-0.823 | 0.001 | | | |
| Sodium | 0.969 | 0.944-0.994 | 0.016 | | | |
| BUN | 1.018 | 1.013-1.023 | 0.000 | | | |
| Bilirubin total | 1.052 | 1.041-1.083 | 0.000 | 1.027 | 1.002-1.052 | 0.037 |
| AST/ALT | 1.704 | 1.482-1.959 | 0.000 | 1.508 | 1.294-1.757 | 0.000 |
| Number of hospital | 0.681 | 0.592-0.784 | 0.000 | | | |
| Cerebrovascular disease | 1.904 | 1.078-3.362 | 0.026 | | | |
| Peptic ulcer disease | 0.415 | 0.193-0.893 | 0.025 | 0.336 | 0.134-0.845 | 0.020 |
| Renal disease | 1.750 | 1.216-2.519 | 0.003 | | | |

[a]*P* value: probability value; [b]RBC: red blood cell; [c]ALT: alanine aminotransferase; [d]ALP: alkaline Phosphatase; [e]AST: aspartate aminotransferase; [f]INR: international normalized ratio; [g]WBC: white blood cell; [h]BUN: blood urea nitrogen

validated model precision, with predicted probabilities closely aligned along the 45 ideal reference line in both cohorts (Figs 3A and 3B).

**Discrimination performance.** The ROC curve analysis demonstrated robust discriminative capacity, with area under the curve (AUC) of 0.846 in the derivation cohort and 0.843 in the validation cohort (Figs 3C and 3D). When benchmarked against conventional scoring systems, our model exhibited superior discriminative ability: derivation cohort AUC improved by 0.015–0.109 compared to MELD (AUC 0.737), MELDs-Na (AUC 0.741), CLIF-OF (AUC 0.817) and CLIF-C ACLF (AUC 0.831), while validation cohort AUC exceeded both scores by 0.059–0.029 (MELD: 0.784; MELD-Na: 0.792; CLIF-OF: 0.813; CLIF-C ACLF: 0.814) (Figs 4A and 4B).

**Clinical utility assessment.** The DCA evaluating threshold probabilities from 20% to 80% demonstrated the model's superior clinical utility. Across this clinically actionable range, the nomogram provided higher net benefit than both "treat-all" and "treat-none" strategies, outperforming MELD, MELD-Na, CLIF-OF and CLIF-C ACLF scores in derivation and validation cohorts (Figs 4C and 4D).. This translates to enhanced capacity for avoiding overtreatment in low-risk patients while appropriately prioritizing high-risk individuals requiring intervention.

**Performance of the model in alcoholic and non-alcoholic liver cirrhosis.** Given that cirrhosis due to alcoholic liver disease may exhibit distinct clinical outcomes compared to other etiologies, we conducted a subgroup analysis. The results indicated no significant difference in short-term mortality or prognosis between patients with alcohol-related cirrhosis and those with non-alcohol-related cirrhosis (*P* = 0.198; S2 Fig). Furthermore, ROC and DCA were performed for both subgroups. The outcomes consistently demonstrated that our model outperformed other models in both discriminatory ability (as measured by AUC) and clinical net benefit (S3 Fig). These findings affirm the robustness and applicability of our predictive model for short-term mortality in patients with cirrhosis, irrespective of alcoholic etiology.

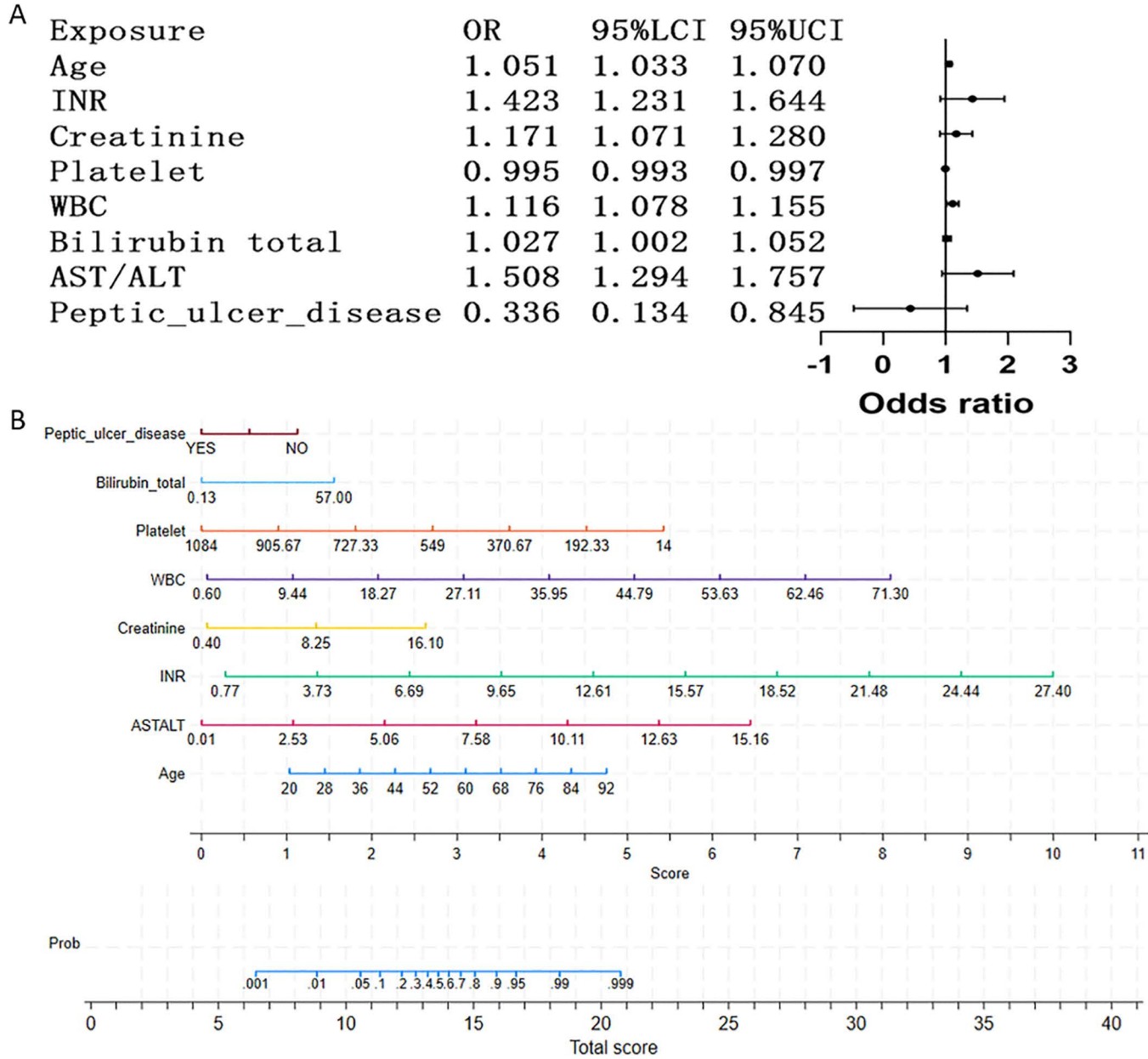

**Fig 2. Multivariable analysis and nomogram development for short-term mortality prediction in severe cirrhosis.** (A) Forest plot of adjusted odds ratios (aORs) derived from multivariable logistic regression. Key predictors include age (aOR 1.051 per year, 95% CI: 1.033–1.070), INR (aOR 1.423, 95% CI: 1.231–1.644), and creatinine (aOR 1.171, 95% CI: 1.071–1.280). All covariates met proportional hazards assumptions (*P* < 0.05). (B) Clinically actionable nomogram integrating significant predictors to estimate individualized 30-day mortality probabilities. Points are assigned per each variable value, with total scores mapped to predicted risk (range: 0.1%–99.9%).

### Short-term prognostic evaluation using cox regression

Univariate analysis identified 19 variables significantly associated with survival time (*P* < 0.05), including laboratory markers (AST, INR, creatinine, hemoglobin, platelets, BUN, WBC, albumin, sodium, total bilirubin, ALP, RBC, and AST/ALT), comorbidities (renal disease, peptic ulcer disease, metastatic solid tumors, cerebrovascular disease), and age.

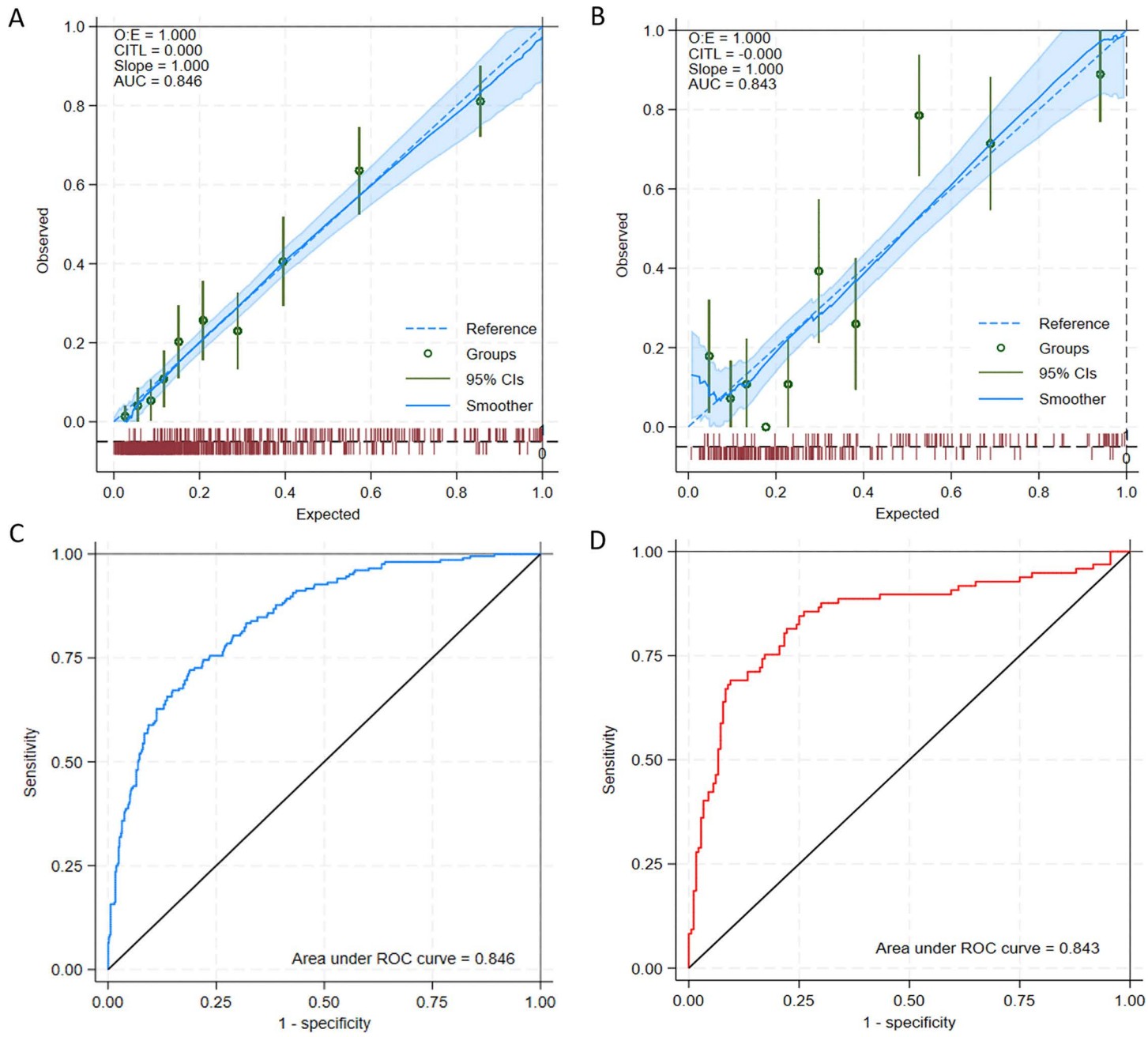

**Fig 3. Libration and discrimination performance of the short-term mortality prediction model in severe cirrhosis.** (A) Calibration curve for the derivation cohort, demonstrating agreement between predicted mortality probabilities (x-axis) and observed outcomes (y-axis). The dashed diagonal represents perfect calibration, while the solid blue line indicates model performance (Brier score<0.25; Hosmer-Lemeshow test $P$>0.05). (B) Validation cohort calibration curve showing preserved accuracy across risk strata (Brier score <0.25; $P$>0.05). (C) Receiver operating characteristic (ROC) curve for the derivation cohort, with the proposed model achieving an AUC of 0.846. (D) External validation ROC curve confirming robust discrimination AUC=0.843.

Multivariable Cox regression (Table 4; Fig 5A) revealed independent predictors across five clinical domains: (1) Hematologic: elevated white blood cell count (adjusted hazard ratio (aHR) 1.046 95% CI 034-1.058 per $10^6$/μL increase) and platelet count (aHR 0.996, 95% CI 0.995–0.998 per ×$10^3$/μL) conferred protective effects; (2) Hepatic: each 1-U/L increase

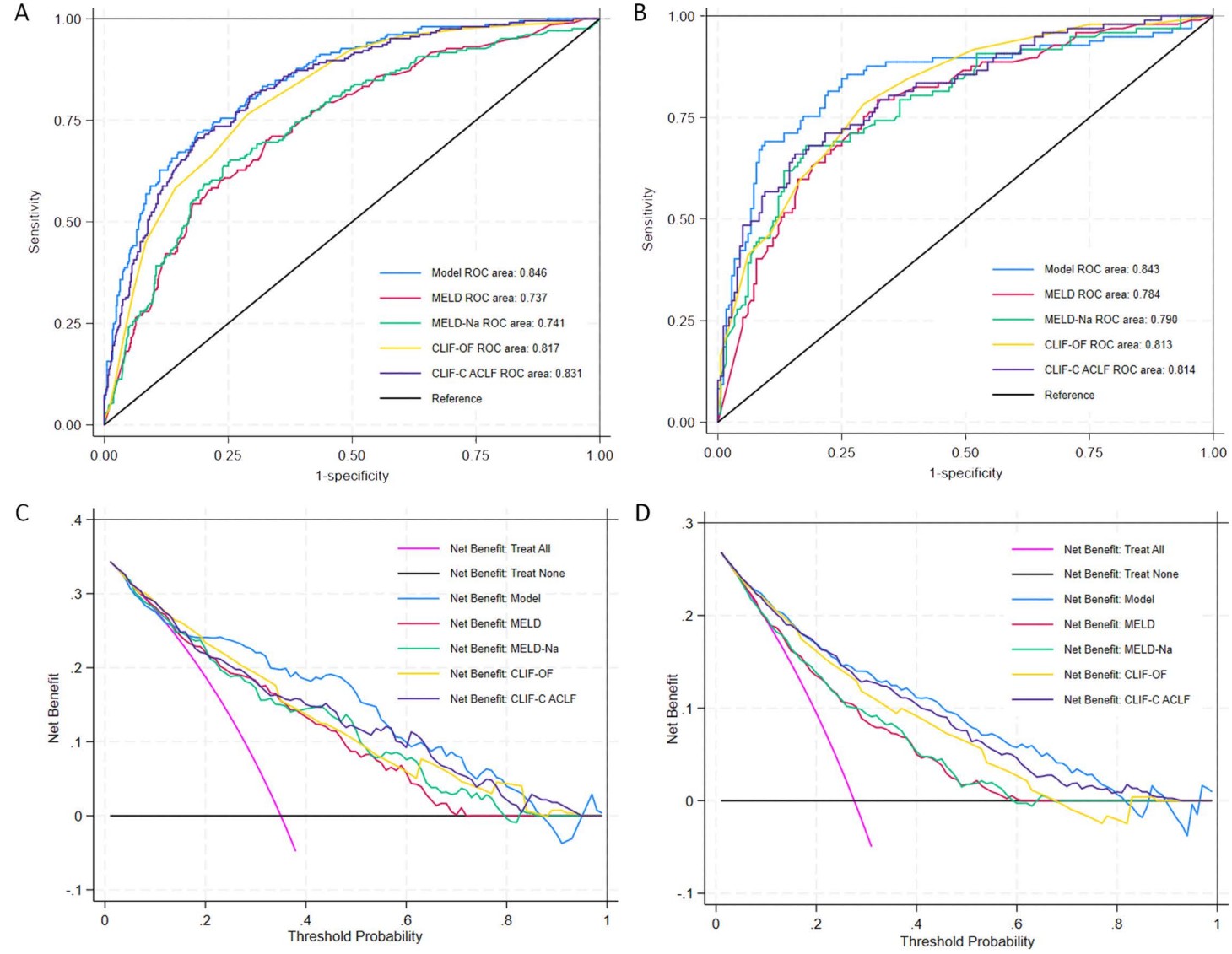

**Fig 4. Nical utility and predictive performance evaluation of the short-term mortality model in severe cirrhosis.** (A) Receiver operating characteristic (ROC) curve comparisons of multiple predictive models in the derivation cohort. The proposed model (red curve) achieved superior discriminative performance (AUC = 0.846) compared to conventional scoring systems. (B) External validation of model performance in an independent cohort, demonstrating maintained predictive accuracy AUC = 0.843. (C) Decision curve analysis (DCA) in the derivation cohort showing enhanced net benefit of the proposed model (blue curve) across clinically relevant threshold probabilities (20–80%), outperforming existing models. (D) Validation cohort DCA confirming consistent clinical utility of the novel model (blue curve) compared to standard approaches.

in AST/ALT (aHR 1.295, 95% CI 1.203–1.393) and 1-mg/dL rise in total bilirubin (aHR 1.023, 95% CI 1.0101.037) amplified mortality risk; (3) Coagulation: INR elevation (aHR 1.088, 95% CI 1.049–1.128 per unit); (4) Renal: 1-mg/dL creatinine increase (aHR 1.102, 95% CI 1.0391.168); (5) Comorbidities: peptic ulcer disease showed paradoxical protection (aHR 0.484, 95% CI 0.2380.987; $P = 0.046$). Bootstrap validation revealed distinct prognostic patterns across covariates. Age, AST/ALT, INR, creatinine, platelet count, WBC, and total bilirubin demonstrated narrow 95% confidence intervals (CIs) excluding the null hazard ratio (HR = 1), indicating statistically significant and precisely estimated effects on short-term prognosis. Conversely, while peptic ulcer disease showed significant associations (CIs excluding HR = 1), their wide

**Table 4. Univariate and multivariate Cox regression analyses.**

| Variables | Univariate Cox Model | | | Multivariate Cox Model | | |
|---|---|---|---|---|---|---|
| | HR | 95%CI | P Value | HR | 95%CI | P Value |
| Age | 1.018 | 1.006-1.029 | 0.002 | 1.034 | 1.022-1.046 | 0.000 |
| alp | 1.002 | 1.001-1.003 | 0.001 | | | |
| AST | 1.000 | 1.000-1.000 | 0.000 | | | |
| INR | 1.130 | 1.098-1.163 | 0.000 | 1.088 | 1.049-1.128 | 0.000 |
| Creatinine | 1.161 | 1.110-1.216 | 0.000 | 1.102 | 1.039-1.168 | 0.001 |
| Hemoglobin | 0.923 | 0.856-0.996 | 0.038 | | | |
| Platelet | 0.999 | 0.997-1.000 | 0.030 | 0.996 | 0.995-0.998 | 0.000 |
| RBC | 0.682 | 0.543-0.855 | 0.001 | | | |
| WBC | 1.048 | 1.038-1.058 | 0.000 | 1.046 | 1.034-1.058 | 0.000 |
| AST/ALT | 1.263 | 1.200-1.329 | 0.000 | 1.295 | 1.203-1.393 | 0.000 |
| Number of hospitalizations | 0.708 | 0.623-0.804 | 0.000 | | | |
| Cerebrovascular disease | 1.700 | 1.061-2.626 | 0.027 | | | |
| Peptic ulcer disease | 0.482 | 0.237-0.978 | 0.043 | 0.484 | 0.238-0.987 | 0.046 |
| Renal disease | 1.583 | 1.169-2.144 | 0.003 | | | |
| Albumin | 0.694 | 0.537-0.899 | 0.006 | | | |
| Sodium | 0.971 | 0.951-0.992 | 0.006 | | | |
| BUN | 1.013 | 1.010-1.016 | 0.000 | | | |
| Bilirubin total | 1.039 | 1.028-1.050 | 0.000 | 1.023 | 1.010-1.037 | 0.001 |

[a]P value: probability value; [b]RBC: red blood cell; [c]ALT: alanine aminotransferase; [d]ALP: alkaline Phosphatase; [e]AST: aspartate aminotransferase; [f]INR: international normalized ratio; [g]WBC: white blood cell; [h]BUN: blood urea nitrogen

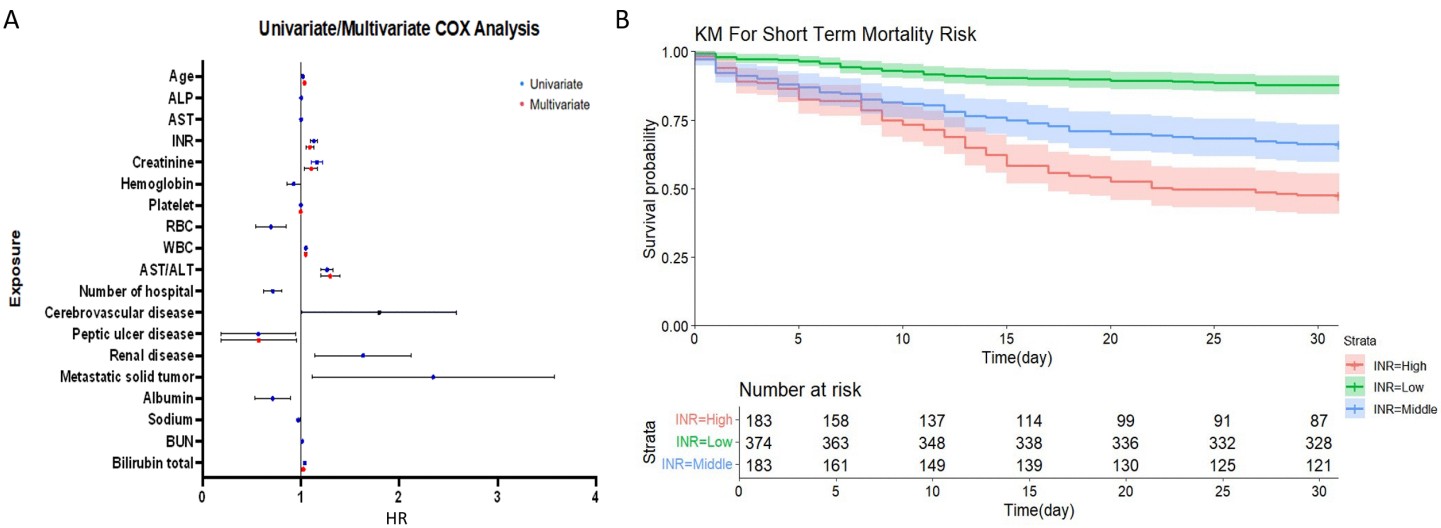

**Fig 5. Prognostic analysis of severe cirrhosis patients.** (A) Forest plot of risk factors associated with 30-day mortality derived from univariate and multivariate Cox regression analyses. Significant predictors (P<0.05) are highlighted, with hazard ratios (HR) and 95% confidence intervals (CI) shown for key variables. (B) Kaplan-Meier survival curves comparing short-term outcomes across distinct INR strata (<1.7, 1.7–2.7, >2.7). Log-rank tests confirmed significant survival differences between groups (P<0.001), with INR>2.7 demonstrating the poorest prognosis.

confidence intervals reflected clinically relevant but imprecisely quantified prognostic effects. Critically, bootstrap-derived HR estimates demonstrated full concordance with original Cox regression results (mean absolute difference <0.05) (S1 Table), validating model stability and supporting clinical utility for mortality risk prediction.

### INR stratification analysis

Using optimal prognostic cutoffs derived from X-tile analysis, we stratified patients into three clinically distinct INR risk categories: low-risk (<1.7, reference), intermediate-risk (1.7–2.7), and high-risk (>2.7). Kaplan-Meier survival curves (Fig 5B) demonstrated significant intergroup disparities (log-rank $P<0.001$), with mortality risk escalating proportionally to INR levels: high-risk (30-day survival 47.4%)> intermediate-risk (66.1%)> low-risk (87.7%). In our validation, such hierarchical differences are also consistent. This graded association underscores INR as a continuous risk modulator in cirrhosis prognosis, providing actionable thresholds for intensity-adjusted monitoring. Internal validation independently confirmed these associations. To further investigate whether the observed grouping could be influenced by anticoagulant therapy, we conducted a subgroup analysis based on anticoagulant use. Kaplan-Meier analysis demonstrated that the INR-based risk stratification remained applicable in both subgroups—anticoagulant users and non-users. No significant difference was observed between the two subgroups (S4 Fig), indicating that the prognostic value of INR stratification was consistent irrespective of anticoagulation status.

## Discussion

### Key findings and model development

This study presents the first comprehensive predictive model for short-term mortality risk in patients with severe cirrhosis, incorporating an analysis of short-term prognosis and evaluating the prognostic significance of the INR. Our findings demonstrate that the prediction model exhibits excellent sensitivity and specificity, with short-term prognosis being significantly associated with multiple factors including age, AST/ALT, INR, creatinine, platelet count, white blood cell count, total bilirubin levels, white blood cell count, and the presence of comorbidities such as peptic ulcer disease. Elevated age, INR, creatinine, white blood cell count, total bilirubin, and AST/ALT ratio were correlated with poorer prognosis and increased short-term mortality risk, whereas decreased platelet levels were also associated with worse outcomes. Interestingly, the presence of peptic ulcer disease was linked to a reduced risk of mortality compared to its absence. Furthermore, significant differences in survival outcomes were observed across different INR ranges, with risk stratification effectively delineated into three distinct intervals. Within these strata, a higher INR was consistently associated with increased short-term mortality risk—regardless of anticoagulant therapy use.

### Portal hypertension plays a central role

Although PHT was not directly included in our model due to database limitations, it emerged as a critical driver in the progression to acute and chronic liver failure, particularly in ACLF. We posit that PHT acts as a common pathway linking multiple observations in our study; it is not merely an isolated hemodynamic issue but a systemic disorder whose effects are mirrored in the spectrum of biomarkers we analyzed. A key finding was that thrombocytopenia—significantly associated with short-term mortality in our cohort—is highly prevalent in cirrhotic patients (reaching 77.9%), and is closely linked to PHT-induced hypersplenism, which promotes platelet sequestration and destruction [28,29]. Similarly, hypersplenism also accelerates red blood cell breakdown. This PHT-related hemolytic process provides a plausible mechanism for the elevated bilirubin observed in our ACLF patients, which has been histopathologically correlated with impaired bile processing and poor prognosis [30]. Furthermore, hemolysis adds to the bilirubin burden, acting as an "accelerator" of hyperbilirubinemia in ACLF [31]. Beyond hematologic effects, PHT compromises intestinal barrier integrity, facilitating bacterial translocation, endotoxemia, and systemic inflammation [32], thereby accounting for the variations in white blood cell counts noted in our analysis. PHT also exacerbates direct liver injury: recent evidence indicates that it promotes hepatic

translocation of pathogens such as Helicobacter pylori (H. pylori), which can induce NLRP3-mediated pyroptosis and elevate aminotransferase levels [33]. Consistently, a study on TIPS showed that reducing portal pressure led to marked improvements in ALT, AST, and INR [34], reinforcing that higher PHT exacerbates hepatocellular damage and dysfunction. Additionally, PHT drives severe extrahepatic complications such as hepatorenal syndrome (suggested by rising creatinine) and peptic ulcer disease through mechanisms like gastric mucosal congestion. In summary, although not directly quantified, the "ghost" of PHT permeates nearly all the biomarkers we studied; it likely interconnects hypersplenism, hepatic decompensation, systemic inflammation, and end-organ damage, serving as a core pathophysiological mechanism in ACLF that is further corroborated by our findings.

## Multidimensional predictors of short-term mortality in severe cirrhosis

The high short-term mortality risk and poor survival prognosis in severe cirrhosis patients underscore the clinical importance of accurate risk prediction. Our analysis revealed that each additional year of age increased short-term mortality risk by 1.04-fold, consistent with existing literature documenting the critical impact of age-related physiological changes on cirrhosis outcomes. These changes include deterioration in circulatory dynamics, organ function, and immune competence, all of which significantly influence the development of cirrhosis-related complications [35]. The strong association between age and prognosis in our study confirms its value as a reliable predictor of short-term mortality in severe cirrhosis.

Renal function emerged as another crucial prognostic factor, with elevated creatinine levels demonstrating significant predictive value. While the precise mechanisms linking cirrhosis to renal impairment remain incompletely understood, our findings support existing research identifying multiple contributing factors: hypovolemia (50% of cases), intrinsic structural kidney damage (30%), hepatorenal syndrome (15–20%), and postrenal obstruction (1%) [36,37]. These diverse pathways collectively impair renal excretory function, leading to creatinine elevation and corresponding increases in mortality risk.

Hematological parameters provided additional prognostic insights. Thrombocytopenia showed an inverse relationship with survival, potentially attributable to both decreased platelet production (mediated through reduced thrombopoietin activity in fibrotic livers [38]) and increased platelet destruction (via hypersplenism and immune-mediated clearance [39,40]). White blood cell count served as an independent predictor of mortality, reflecting the high prevalence (24–29%) of bacterial infections in advanced cirrhosis and their substantial contribution to poor outcomes [41,42]. The inclusion of this parameter addresses a notable gap in previous prediction models that overlooked infection-related mortality.

Total bilirubin emerged as a significant predictor of short-term mortality in severe cirrhosis, a finding supported by its odds ratio (OR 1.027, 95% CI 1.002–1.052) in our model and consistent prognostic value in survival analysis (HR 1.023, 95% CI 1.010–1.037, p = 0.001). The elevation in total bilirubin reflects both hepatic and extrahepatic pathophysiological processes. On one hand, portal hypertension-induced hypersplenism promotes chronic hemolysis through splenic sequestration and destruction of red blood cells, exacerbated by altered hemorheological properties and microvascular stress [43,44]. This leads to increased indirect bilirubin from hemoglobin breakdown. On the other hand, intrahepatic cholestasis and impaired hepatocyte function reduce bilirubin clearance, elevating direct bilirubin levels [45]. Additionally, in decompensated cirrhosis, diminished glucuronidation capacity further contributes to the accumulation of indirect bilirubin. Together, these mechanisms underpin the strong association between elevated total bilirubin and mortality, highlighting its utility in risk stratification and early intervention for high-risk cirrhotic patients [46,47].

In our predictive model, we observed that critically ill cirrhotic patients with peptic ulcer disease exhibited a lower short-term mortality compared to those without peptic ulcers. This finding appears to contradict certain studies indicating an increased risk of death within 90 days after hospitalization for peptic ulcer in cirrhotic patients [48]. The discrepancy may be attributed to several factors. Firstly, the subgroup of patients with peptic ulcers might have had less severe overall disease. Compared to complications such as esophagogastric variceal bleeding, peptic ulcer hemorrhage is associated with a lower risk of mortality, which could largely explain our results. Secondly, cirrhotic patients diagnosed with peptic ulcers likely received comprehensive therapeutic regimens, including non-steroidal anti-inflammatory drugs, mucosal protectants,

proton pump inhibitors, acid suppressants, antibiotics, prophylactic low-dose aspirin, antifibrinolytic agents, as well as hemostatic and transfusion therapies for bleeding ulcers. These interventions may have reduced the risk of hemorrhage and other complications in severe cirrhosis, thereby potentially contributing to a lower short-term mortality in this patient group [49,50].

The INR is an established predictor of bleeding in patients with liver and acute diseases [51]. Our analysis confirms that INR levels also effectively predict short-term mortality in critically ill cirrhotic patients, with significant disparities across INR strata. This association stems from the liver's central role in synthesizing coagulation proteins—such as albumin, proteins C and S, and clotting factors—whose production is impaired in ACLF, leading to elevated INR [52,53]. Concurrent organ failures, such as acute kidney injury, may exacerbate coagulation dysfunction through metabolic disturbances and impaired toxin clearance. Although studies like that of Amber Afzal et al suggest that INR-associated bleeding risk varies with etiology and medication (e.g.,warfarin use leading to exponential risk increase above INR [54]), our subgroup analyses demonstrated that INR stratification consistently discriminated short-term mortality risk regardless of anticoagulant use. Higher INR levels correlated with increased bleeding risk and worse prognosis, consistent with existing literature.

## Methodological innovations

This study is distinguished by three key advancements: the systematic integration of comorbidities, addressing oversights in previous models; the use of extreme-value laboratory analysis to capture critical physiological states more accurately than averaged measures; the range between maximum and minimum values may cause the mean to inaccurately reflect the patient's true clinical status. To mitigate this bias, we employed extreme analysis, which more reliably captures disease severity during hospitalization, although the use of extreme values may introduce bias by incorporating post-admission information. A post-hoc sensitivity analysis demonstrated stable predictive performance when only initial admission values were used (Training AUC: 0.771; Validation AUC: 0.767), thereby supporting the robustness of the model (S5 Fig); and the introduction of clinically actionable INR stratification, replacing traditional binary classifications. These innovations yielded a model demonstrating superior discrimination compared to existing scores while maintaining practicality through the use of routine parameters.

## Study limitations

**Data source and generalizability.** First, all data were sourced solely from the MIMIC-IV database, a single-center repository; generalizability to other regions or populations requires external validation. Second, our cohort consisted exclusively of cirrhotic patients admitted to the ICU, limiting the applicability of our findings to non-ICU settings.

**Data handling and missing values.** Regarding data processing, variables with missing values exceeding 20% were excluded, which may have omitted potentially significant predictors. For variables with less than 20% missingness, imputation was applied, possibly introducing bias.

**Methodological approach.** Methodologically, our analysis relied solely on logistic regression for prediction modeling and Cox regression for prognostic analysis. The use of alternative machine learning or statistical approaches might enhance model performance.

## Future directions

In the future, we will first validate and refine our model through multicenter studies and in non-ICU clinical settings to enhance its generalizability and accuracy. Subsequently, an initial version of this predictive model has been deployed as an online mortality risk calculator, accessible at: https://9zhangshun.shinyapps.io/livercirrhosis/. Future work will focus on further refining the tool's performance and predictive accuracy. Our long-term goal is to develop a comprehensive web-based application that can be seamlessly integrated into routine clinical workflows. We expect that these efforts will contribute to improved global health outcomes and support clinical decision-making for patients with severe cirrhosis.

## Conclusions

This study establishes a clinically implementable prediction model for short-term mortality in severe cirrhosis, demonstrating superior performance to traditional prognostic scores through innovative parameter selection and INR stratification. While our findings provide critical insights into comorbidity-associated risk modulation and coagulation parameter interpretation, prospective validation across diverse care settings remains essential prior to clinical implementation. Future research directions should prioritize integration of dynamic laboratory trends and therapeutic response biomarkers to enhance predictive precision.

## Supporting information

**S1 Fig. Variable selection with LASSO.** (A) LASSO cross-validation error curve. The number of variables selected at the optimal lambda value is 8. (B) LASSO coefficient plot. Variables with non-zero coefficients (marked "Selected: Yes") are retained, identifying key variables.
(TIF)

**S2 Fig. Relationship between alcoholic liver disease, non-alcoholic liver disease and short-term mortality.** In the subgroup analysis, there were 545 patients in the alcoholic cirrhosis group, among whom 161 (29.5%) experienced short-term mortality and 384 (70.5%) survived in the short term. In the non-alcoholic cirrhosis group, there were 499 patients, with 145 (29.1%) experiencing short-term mortality and 354 (70.9%) surviving in the short term. Statistical analysis showed no significant difference in short-term mortality between the two groups ($P = 0.918$).
(TIF)

**S3 Fig. Comparative Evaluation of Predictive Models and Clinical Benefit Analysis for Short-Term Mortality in Patients with Severe Alcoholic vs. Non-Alcoholic Cirrhosis.** (A) Comparison of AUC values among multiple models in the alcoholic cirrhosis group. (B) Comparison of AUC values among multiple models in the non-alcoholic cirrhosis group. (C) Clinical decision curve analysis for multiple models in the alcoholic cirrhosis group. (D) Clinical decision curve analysis for multiple models in the non-alcoholic cirrhosis group.
(TIF)

**S4 Fig. Kaplan-Meier Analysis by Anticoagulant Use.** (A) Anticoagulant Group: Short-term survival in severe cirrhosis stratified by international normalized ratio (INR) of prothrombin. (B) Non-Anticoagulant Group: Short-term survival in severe cirrhosis stratified by INR of prothrombin.
(TIF)

**S5 Fig. Sensitivity analysis of the prediction model.** (A) ROC curves of the model in the training cohort. The blue curve represents the model using extreme laboratory values (AUC = 0.837), and the red curve represents the model using the first recorded values at admission (AUC = 0.771). (B) ROC curves of the model in the validation cohort. The blue curve corresponds to the model using extreme laboratory values (AUC = 0.825), and the red curve corresponds to the model using the first recorded values at admission (AUC = 0.767).
(TIF)

**S1 Table. Bootstrap test for Cox regression.**
(DOCX)

**S1 File. Raw data.**
(XLSX)

## Acknowledgments

We acknowledge the MIMIC dataset for offering diverse data resources, allowing us to pursue comprehensive and profound research.

## Author contributions

**Conceptualization:** Shun Zhang, Rui Liu, Tao Pan, Xudong Wen.

**Data curation:** Shun Zhang, Rui Liu, Zhengjie Li.

**Formal analysis:** Shun Zhang, Zhengjie Li, Tao Pan.

**Funding acquisition:** Xudong Wen.

**Methodology:** Shun Zhang, Rui Liu.

**Project administration:** Shun Zhang.

**Resources:** Zhengjie Li, Xudong Wen.

**Software:** Rui Liu, Zhengjie Li.

**Supervision:** Shun Zhang, Rui Liu, Zhengjie Li, Tao Pan, Xudong Wen.

**Validation:** Shun Zhang, Zhengjie Li.

**Visualization:** Zhengjie Li, Tao Pan.

**Writing – original draft:** Shun Zhang, Rui Liu.

**Writing – review & editing:** Tao Pan, Xudong Wen.

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
