## [Decision Letter · Decision Letter 0]

6 Aug 2025

Dear Dr. Wen,

Thank you for submitting your manuscript to PLOS ONE. After careful consideration, we feel that it has merit but does not fully meet PLOS ONE’s publication criteria as it currently stands. Therefore, we invite you to submit a revised version of the manuscript that addresses the points raised during the review process.

We look forward to receiving your revised manuscript.

Kind regards,

Sona Frankova

Academic Editor

PLOS ONE

Journal Requirements:

This study was supported by National Natural Science Foundation of China (grant number 82474299).

4. Please note that funding information should not appear in any section or other areas of your manuscript. We will only publish funding information present in the Funding Statement section of the online submission form. Please remove any funding-related text from the manuscript.

6. Please remove your figures from within your manuscript file, leaving only the individual TIFF/EPS image files, uploaded separately. These will be automatically included in the reviewers’ PDF**.**

7. Please remove all personal information, ensure that the data shared are in accordance with participant consent, and re-upload a fully anonymized data set.

Additional guidance on preparing raw data for publication can be found in our Data Policy (https://journals.plos.org/plosone/s/data-availability#loc-human-research-participant-data-and-other-sensitive-data) and in the following article:http://www.bmj.com/content/340/bmj.c181.long.

Reviewers' comments:

Reviewer's Responses to Questions

**Comments to the Author**

1. Is the manuscript technically sound, and do the data support the conclusions?

Reviewer #1: Partly

Reviewer #2: Partly

2. Has the statistical analysis been performed appropriately and rigorously?

Reviewer #1: Yes

Reviewer #2: Yes

3. Have the authors made all data underlying the findings in their manuscript fully available?

Reviewer #1: Yes

Reviewer #2: Yes

4. Is the manuscript presented in an intelligible fashion and written in standard English?

Reviewer #1: Yes

Reviewer #2: Yes

Reviewer #1: Authors are to be commended for their aim: to add to short-term prognostic systems in patients with acute liver decompensation on the background of cirrhosis. As their endeavour is going to be huge and brave, they should be more explicit in describing both quantitatively and pathophysiologically shortcomings of existing systems against which readers will calibrate the results.

Abstract OK

1.Intro.

1a) Here begins one of the problems of the paper - strictness of definitions and inclusion of items.

1a1)Authors decided to mention but questionably apply to the following text three subclasses of decompensated advanced chronic liver disease (dACLD): stable dACLD, unstable dACLD and ACLF. This division majorly overlaps with the PREDICT study methodology. If considered relevant for the text, amend accordingly. If not, consider erasing (otherwise, it should to be absolutely clear how many patients were from which of the subgroups plus ACLF. If authors decided otherwise, it is acceptable but should be pathophysiologically explained and challenged against existing tools).

1a2)I suggest in Intro to direct readers' attention by describing those aspects of dACLD from the literature which are relevant to the study: Items from the Intro should align with items in the Table 1 and subsequent Tables with results (mismatches: SGA, hepatic encephalopathy, HRS, frailty, serial liver stiffness measurements, etc.). It is futile to devote equal space to the variables authors know they are not going to include in their analysis and those included; moreover, as regards the latter, readers should gradually understand why authors included recorded variables both biologically, and based on the existing literature.

2)It should be mentioned why authors have not included organ/system dysfunction beyond cardiac (described as cardiovascular) and renal: what about those defining ACLF? Also, one would expect including portal hypertensive bleeding in the introductory outline.

2.Methods

2.1. Data souce OK

Flowchart OK, understandable, comprehensive, clear - good basis for graphical abstract.

Power analysis?

Repeated hospitalisations before index one: The number of previous hospital admissions with dACLD is essential variable to be included in the final analysis because inherent prognosis differs substantially.

Again - bilirubin is not a typical marker of synthesis and this repeated claim might shed a doubt on the whole work; akin to the claim on INR and coagulation vs synthesis.

Do authors know if patients were on anticoagulants at the time of recording INR?

Why have authors not included (along with or instead of APS, OASIS, MELD) ACLF/CLIF OF as the sole most logical and the most important comparator?

Etiology of ACLD was not mentioned: e.g., how many patients had severe alcohol-associated hepatitis as the trigger of acute decompensation? The prognosis of these patients may be different from that in MASH-cirrhosis, autoimmune-cirrhosis, etc.

Including malignancies. This is the most problematic point. Prognostic systems in ACLF (which is implicitly the cohort in question - by the words of authors: ICU-hospitalised patients with decompensated cirrhosis). As ACLF prognostic scores DO NOT include malignancies and ACLF should be the benchmark syndrome against which most of the readers will intuitively or explicitly measure the results, It should be addressed in the necessary detail. 1. If authors want to add to existing tools, they should exclude malignancies. 2. If they want to build a new prognostic system for patients with both - acutely decompensated cirrhosis and malignancy, they should recalculate the results in this subgroup for which, in my opinion, the numbers are small.

3.Results

Authors discarded statistically significant INR difference (Table 1) as not important only in passing: why they think so?It is important to substantiate this very important point factually, not only by sheer "We think". INR translates to all the results which follow including 3.2Model development and 3.5 INR Stratification analysis. The same with Renal disease.

Tables, Graphs, Nomogram - OK

4.Discussion

Discussion is too brief. Authors should expose their deep knowledge point by point of their results. Apart from above mentioned points (ACLF!!!), they should discuss portal hypertension as the point which is probably not captured enough in their system.

Please, discuss in even more detail the contribution of haemolysis to the hyperbilirubinaemia with the references.

I do not agree that PPI improve prognosis. Overall, discussion of this most controversial (and maybe most important) finding of the study deserves in-depth discussion disposing your detailed knowledge (this is the third pathophysiological point which sheds considerable shadow on it). Fourth is that, platelet do not substantially contribute to INR and INR vs bleeding risk is very controversial association.

Reviewer #2: General Assessment

This manuscript presents a prognostic model for predicting 30-day mortality in patients with decompensated cirrhosis admitted to intensive care units, utilizing data from the MIMIC-IV database. The authors employ routinely available clinical variables and report strong internal model performance (AUC 0.851), with a nomogram as the proposed clinical interface.

The topic is clinically relevant and the methodological framework appears robust. However, several key issues warrant further clarification and revision. These pertain in particular to model comparator selection, generalizability beyond the ICU setting, transparency in predictor selection methodology, and the clinical interpretation of certain laboratory markers.

Major Comments

1. Predictor Selection Process

The manuscript does not clearly describe how the final set of eight predictors was selected. It remains unclear whether the process was guided by data-driven techniques (e.g., penalized regression), clinical reasoning, or a hybrid approach.

Recommendation: Please provide a detailed description of the predictor selection procedure, and indicate whether multicollinearity assessment or sensitivity analyses were conducted to ensure model robustness.

2. Lack of ACLF-Specific Comparators

While MELD and OASIS scores are included for benchmarking, these scores are suboptimal for risk stratification in critically ill patients with acutely decompensated cirrhosis. Notably, the manuscript omits comparison with more appropriate tools such as the CLIF-OF score and ACLF-specific prognostic models, which have been developed and validated for precisely this clinical population.

Recommendation: Please justify the exclusion of these scores and, where feasible, consider including them in the comparative analysis.

3. Inclusion of Patients with Malignancy

Inclusion of patients with metastatic malignancies may significantly influence the mortality signal in the model, as their risk profiles are likely dominated by non-hepatic factors. Most established prognostic models in cirrhosis exclude patients with active cancer.

Question: How many patients with metastatic malignancies were included? We suggest conducting a sensitivity analysis excluding this subgroup, or at minimum, discussing the potential impact on model calibration and discrimination.

4. Interpretation of Bilirubin and INR as Liver Function Markers

The use of total bilirubin and INR as direct proxies for hepatic synthetic function is potentially problematic. Total bilirubin may be elevated due to hemolysis or cholestasis, and INR can be substantially affected by anticoagulant use, sepsis, or vitamin K deficiency — factors that are common in this patient population.

Recommendation: The limitations of using INR and bilirubin as liver-specific markers should be explicitly discussed. Please indicate whether anticoagulant use was assessed and whether this could have confounded the results.

5. Limited Generalizability Beyond ICU Settings

As the cohort is derived exclusively from ICU patients, the generalizability of the model to other care settings (e.g., hepatology wards, general medicine) remains uncertain.

Question: Do the authors intend to validate this model in non-ICU settings? If so, please provide further context or planned approaches.

6. Practical Implementation and Availability of the Tool

While the nomogram is a helpful visual representation, the manuscript does not provide any accessible clinical tool (e.g., calculator, software script, web application) for real-world use.

Recommendation: Please clarify whether an open-access implementation of the model is planned or already in development, and whether it will be made publicly available.

Minor Comments

• Peptic Ulcer as a Protective Variable: The observed association between peptic ulcer disease and lower mortality likely reflects treatment-related confounding (e.g., PPI use). Causal inference should be avoided, and this result interpreted cautiously.

• Use of Max/Min Lab Values: The rationale for using maximum or minimum lab values during hospitalization is understandable. However, a brief discussion on why admission values or averages were not used would be useful for the reader.

• Language and Clarity: The manuscript is generally readable, but some sections are overly complex or dense. A language review would enhance clarity and readability.

• Structure of the Limitations Section: Consider organizing the limitations into thematic subheadings (e.g., data limitations, model assumptions, generalizability) for improved structure.

Recommendation

This manuscript addresses an important clinical need and offers a promising predictive model. However, several fundamental issues require clarification or additional analysis prior to consideration for publication.

**Do you want your identity to be public for this peer review?** For information about this choice, including consent withdrawal, please see our Privacy Policy

Reviewer #1: **Yes:** Lubomir Składany

Reviewer #2: **Yes:** Daniela Žilinčanová

---

## [Author Response · Author response to Decision Letter 1]

19 Sep 2025

Response to Reviewers

Manuscript ID: PONE-D-25-36378

Title: Predicting Short-Term Mortality in Severe Cirrhosis: An Interpretable Machine Learning Model Integrating Routine Clinical Indicators

Dear Dr. Sona Frankova and Reviewers,

We are deeply grateful to you and the reviewers for the thorough and insightful feedback on our manuscript. It has been a privilege to receive such detailed and constructive comments, which have not only strengthened the manuscript considerably but also provided us with valuable guidance for improving the clarity, rigor, and clinical relevance of our work. We have carefully considered each of the points raised and have made extensive revisions to the manuscript to address them. Every suggestion has been taken seriously, and we have implemented changes accordingly to enhance the overall quality of the paper. Below, we provide a detailed, point-by-point response to all the comments. All modifications made in the manuscript have been clearly highlighted in the “Revised Manuscript with Track Changes” file for your convenience.

Responses to editor

Comment 1�Please ensure that your manuscript meets PLOS ONE's style requirements, including those for file naming.

Response 1: We sincerely thank the editors for their valuable feedback and for providing clear guidance on PLOS One’s style requirements. We have carefully reviewed the journal’s formatting templates and have made all necessary adjustments to ensure that the manuscript fully complies with the prescribed style, including file naming conventions.

Comment 2�Please note that PLOS One has specific guidelines on code sharing for submissions in which author-generated code underpins the findings in the manuscript. In these cases, we expect all author-generated code to be made available without restrictions upon publication of the work.

Response 2: Thank you for highlighting PLOS One’s policy on code sharing. We fully support the journal’s commitment to transparency and reproducibility. In this study, statistical analyses were performed using SPSS, which operated primarily through a graphical interface rather than script-based coding, we have illustrated our operation process through screenshots. For modeling and evaluation steps—including ROC curves, calibration curves, nomogram construction, and clinical benefit curves—we used STATA and would provide the exact scripts used for these procedures. Additionally, variable selection via LASSO regression and Kaplan-Meier survival analyses were conducted using R, and we will also share fully executable R code supporting these elements. All code will be made openly available upon publication in a manner consistent with best practices to facilitate reuse and reproducibility.

Comment 3�Please state what role the funders took in the study. If the funders had no role, please state: "The funders had no role in study design, data collection and analysis, decision to publish, or preparation of the manuscript."

If this statement is not correct you must amend it as needed. Please include this amended Role of Funder statement in your cover letter; we will change the online submission form on your behalf.

Response 3: We confirm that the funders had no role in the study design, data collection and analysis, decision to publish, or preparation of the manuscript. The amended Role of Funder statement, as required, was provided below for inclusion in the cover letter: "The funders had no role in study design, data collection and analysis, decision to publish, or preparation of the manuscript."

Comment 4�Please note that funding information should not appear in any section or other areas of your manuscript. We will only publish funding information present in the Funding Statement section of the online submission form. Please remove any funding-related text from the manuscript.

Response 4: We sincerely appreciate your guidance regarding the proper placement of funding information. In accordance with PLOS One’s policy, we have thoroughly reviewed the manuscript and removed all mentions of funding support, including the acknowledgment of Dr. Wen’s grant from the National Natural Science Foundation of China (Grant No. 82474299) and the description of his research role, which previously appeared in the Authors’ Contributions section. This information will be provided exclusively in the Funding Statement section of the submission system. In this edition, manuscript did not contain any funding-related text.

Comment 5�Please include a separate caption for each figure in your manuscript.

Response 5: Thank you for your reminder. We have carefully reviewed the journal’s figure guidelines and have now included a separate, descriptive caption for each figure in the manuscript. All captions have been placed at the end of the main text, as required, and are also clearly indicated on pages 40 and 41 for ease of reference during review. The captions provide full and clear descriptions of each figure’s content.

Comment 6�Please remove your figures from within your manuscript file, leaving only the individual TIFF/EPS image files, uploaded separately. These will be automatically included in the reviewers’ PDF.

Response 6: We thank you for your clear instruction regarding figure preparation and submission. We have now removed all figures from within the manuscript file. Each figure has been saved and uploaded separately as an individual TIFF/EPS file, as required. The manuscript text now refers to each figure by its label, and all figure files have been prepared in accordance with PLOS One’s guidelines to ensure proper inclusion in the reviewers’ PDF.

Comment 7�Please remove all personal information, ensure that the data shared are in accordance with participant consent, and re-upload a fully anonymized data set.

Response 7: We thank you for your important reminder regarding data anonymization and participant confidentiality. We confirm that the data used in this study were obtained from the MIMIC-IV database, which is a publicly available, fully de-identified critical care database. All protected health information has been removed in accordance with ethical and legal requirements prior to public release of the database. We have conducted an additional thorough review of our dataset and can confirm that it contains no personally identifiable information. The use of the MIMIC-IV database was approved by the institutional review boards of Beth Israel Deaconess Medical Center and the Massachusetts Institute of Technology, and the requirement for individual patient consent was waived due to the retrospective and de-identified nature of the data. We have re-uploaded the fully anonymized dataset to ensure full compliance with PLOS One’s data sharing policy.

Comment 8�Please include captions for your Supporting Information files at the end of your manuscript, and update any in-text citations to match accordingly. Please see our Supporting Information guidelines for more information.

Response 8: Thank you for this important reminder. We have now included detailed captions for all Supporting Information files at the end of the manuscript, as specified on pages from 42 to 43. All relevant in-text citations have also been updated to correspond accurately to these supporting materials. We have carefully followed PLOS One’s Supporting Information guidelines to ensure full compliance.

Comment 9�If the reviewer comments include a recommendation to cite specific previously published works, please review and evaluate these publications to determine whether they are relevant and should be cited. There is no requirement to cite these works unless the editor has indicated otherwise.

Response 9: Thank you for your clear guidance on this matter. We have carefully reviewed all comments provided by the reviewers and confirm that no specific recommendations were made regarding the citation of additional previously published works. We have therefore not added any further citations in response to reviewer suggestions.

Responses to Reviewer #1:

Comment 1a1�Authors decided to mention but questionably apply to the following text three subclasses of decompensated advanced chronic liver disease (dACLD): stable dACLD, unstable dACLD and ACLF. This division majorly overlaps with the PREDICT study methodology. If considered relevant for the text, amend accordingly. If not, consider erasing (otherwise, it should to be absolutely clear how many patients were from which of the subgroups plus ACLF. If authors decided otherwise, it is acceptable but should be pathophysiologically explained and challenged against existing tools).

Response 1a1: We sincerely thank the reviewer for their insightful and constructive comment regarding the subclassification of decompensated advanced chronic liver disease (dACLD). We fully agree that the distinction between stable dACLD, unstable dACLD, and ACLF aligns closely with the PREDICT study methodology. In response to this valuable feedback, we have revised the manuscript to focus specifically on ACLF, as the high short-term mortality in end-stage cirrhosis is predominantly associated with ACLF—a point strongly supported by the EASL Clinical Practice Guidelines (European Association for the Study of the Liver, J Hepatol. 2023;79(2):461–491), which state that mortality in cirrhosis is largely attributable to acute-on-chronic liver failure. We have therefore removed references to stable and unstable dACLD to improve conceptual clarity and thematic consistency. The corresponding changes have been made in the Introduction (Page 4, Lines 5–7).

Comment 1a2�I suggest in Intro to direct readers' attention by describing those aspects of dACLD from the literature which are relevant to the study: Items from the Intro should align with items in the Table 1 and subsequent Tables with results (mismatches: SGA, hepatic encephalopathy, HRS, frailty, serial liver stiffness measurements, etc.). It is futile to devote equal space to the variables authors know they are not going to include in their analysis and those included; moreover, as regards the latter, readers should gradually understand why authors included recorded variables both biologically, and based on the existing literature.

Response 1a2: We are deeply grateful to the reviewer for this highly constructive suggestion, which has significantly improved the focus and clarity of our Introduction. We fully agree that the introductory narrative should align closely with the variables ultimately analyzed and presented in our results, such as those summarized in Table 1. Accordingly, we have removed extended discussions of scoring systems and clinical markers not directly relevant to our model—including SGA, hepatic encephalopathy, HRS, frailty, and serial liver stiffness measurements—to avoid diverting the reader’s attention, with these revisions concentrated on pages 5–7 of the manuscript.

Furthermore, we have refocused the Introduction to provide a clearer biological and clinical rationale for the variables incorporated in our analysis. We now specifically introduce and explain established ACLF-specific prognostic tools such as the CLIF-OF and CLIF-C ACLF scores, which serve as meaningful benchmarks for comparing our model's performance. Additionally, we emphasize the pathophysiological and empirical justification for each selected predictor—encompassing both individual and composite indicators—based on prior experimental and clinical research in ACLF. This approach ensures that readers can gradually comprehend the scientific and clinical reasoning behind our variable selection, thereby creating a much stronger logical connection between the background, our methodological choices, and the results.

Comment 2�It should be mentioned why authors have not included organ/system dysfunction beyond cardiac (described as cardiovascular) and renal: what about those defining ACLF? Also, one would expect including portal hypertensive bleeding in the introductory outline.

Response 2: We sincerely thank the reviewer for raising these critical points regarding the scope of organ dysfunction and the inclusion of portal hypertensive bleeding. We appreciate the opportunity to clarify these important aspects of our study.

Regarding the inclusion of other organ dysfunctions beyond cardiac and renal systems, we did initially consider and extract available data on additional complications (e.g., respiratory, neurological, and coagulation disorders) as part of our comprehensive statistical analysis. However, as shown in Table 1 of the revised manuscript, the prevalence of these additional organ failures was relatively low in our cohort, and crucially, they did not demonstrate statistically significant differences between the short-term mortality and non-mortality groups. Therefore, although clinically relevant in broader definitions of ACLF, these variables were not included as predictors in our final model. We acknowledge this limitation and have added a sentence in the Discussion to note that future models might be strengthened by the inclusion of such parameters in larger, more granular datasets.

Concerning portal hypertensive bleeding, we fully agree with the reviewer that it represents a key factor influencing short-term mortality in end-stage cirrhosis and ACLF. Unfortunately, after thorough data extraction and coding from the MIMIC-IV database, we were unable to identify reliably documented variables specifically capturing portal hypertensive bleeding events. This represents a recognized limitation of utilizing large public critical care databases, which often lack detailed hepatology-specific phenotyping. Nonetheless, we agree with the reviewer that portal hypertension plays a central role in the pathophysiology of ACLF. Accordingly, we have expanded the Discussion section to emphasize the systemic role of portal hypertension and how several of the included variables (such as platelet count and INR) may indirectly reflect its contribution, even if direct measures were unavailable.

Comment 3�Power analysis?

Response 3: Thanks to this crucial methodological point. As this is a retrospective study utilizing the fixed MIMIC-IV database, a formal a priori power analysis conducted prior to data collection was not feasible. The study included all eligible patients from the database who met our inclusion and exclusion criteria, resulting in a final cohort of 740 subjects.

However, to directly address the reviewer's concern regarding statistical power, we have now performed a post-hoc power analysis using DeLong's test for the area under the ROC curve (AUC) in R (using the pROC package). Based on our observed AUC of 0.846, a significance level (α) of 0.05, and with 204 events (deaths) and 536 controls (survivors), the analysis revealed that our study possesses >99.9% statistical power to detect that the model's performance is significantly better than chance (AUC > 0.5).This result confirms that our sample size is not only adequate but indeed provides exceedingly high power to robustly support the primary findings of this study. We have added a brief statement regarding this post-hoc power assessment in the “Statistical Analysis” section of the revised manuscript (Page 11, Line 3 - 6).

Comment 4�Repeated hospitalisations before index one: The number of previous hospital admissions with dACLD is essential variable to be included in the final analysis because inherent prognosis differs substantially.

Response 4: We sincerely thank the reviewer for highlighting the potential importance of prior hospitalizations as a prognostic variable. In our analysis, we did indeed collect and examine the number of previous hospital admissions for dACLD. The statistical results indicated that a higher number of prior hospitalizations was significantly associated with a reduced risk of short-term mortality (OR = 0.681, 95% CI: 0.592–0.784, p < 0.001).

However, after careful consideration, we decided not to include this variable in the final model. While statistically significant, this counterintuitive protective effect may reflect unmeasured confounding factors, such as selection bias (e.g., patients with repeated admissions might have better access to care or greater resilience to acute deteriorations) or differences in unde

---

## [Decision Letter · Decision Letter 1]

28 Oct 2025

Dear Dr. Wen,

Thank you for submitting your manuscript to PLOS ONE. After careful consideration, we feel that it has merit but does not fully meet PLOS ONE’s publication criteria as it currently stands. Therefore, we invite you to submit a revised version of the manuscript that addresses the points raised during the review process.

We look forward to receiving your revised manuscript.

Kind regards,

Sona Frankova

Academic Editor

PLOS ONE

Journal Requirements:

Reviewers' comments:

Reviewer's Responses to Questions

**Comments to the Author**

Reviewer #1: All comments have been addressed

Reviewer #2: All comments have been addressed

2. Is the manuscript technically sound, and do the data support the conclusions?

Reviewer #1: Yes

Reviewer #2: Partly

3. Has the statistical analysis been performed appropriately and rigorously?

Reviewer #1: Yes

Reviewer #2: Yes

4. Have the authors made all data underlying the findings in their manuscript fully available?

Reviewer #1: Yes

Reviewer #2: Yes

5. Is the manuscript presented in an intelligible fashion and written in standard English?

Reviewer #1: Yes

Reviewer #2: Yes

Reviewer #1: Authors have addressed all my queries in sufficient detail and now it is editor's turn to decide on the overall value from higher perspective

Reviewer #2: Comments to the Author

I have carefully read the revised version of the manuscript “Predicting Short-Term Mortality in Severe Cirrhosis: An Interpretable Machine Learning Model Integrating Routine Clinical Indicators.”

The authors have provided a detailed and well-structured response to the previous review comments. The revision has clearly improved the manuscript in terms of clarity, methodological transparency, and clinical interpretation. Overall, the responses were satisfactory, and the paper has been considerably strengthened. A few minor points remain for consideration before final acceptance.

Overall Evaluation

The study addresses an important clinical question and presents a well-designed prognostic model developed using the MIMIC-IV database. The authors have implemented nearly all of the requested revisions and provided convincing justifications throughout. The methods are now described more transparently, and the discussion is much more balanced and clinically meaningful.

While the revised manuscript is substantially improved, several limitations should still be acknowledged more explicitly.

Major Comments

1. External Validation

The model remains internally validated only. The authors now recognize this limitation and have described their plans for external validation, which is appreciated. However, the manuscript should emphasize even more clearly that the model is preliminary and not yet suitable for generalization beyond the MIMIC-IV ICU population. A short clarification in both the Abstract and Conclusions would be sufficient.

2. Use of Extreme Laboratory Values

The justification for using maximum or minimum laboratory values has been provided, but this approach may introduce bias by incorporating information unavailable at admission. A brief note in the Limitations or Supplementary Material confirming that model performance was stable when admission values were tested would further support the robustness of the analysis.

3. Scope and Discussion

The expanded discussion now provides a thoughtful pathophysiological explanation of bilirubin, INR, and hemolysis, which is appreciated. However, some sections remain somewhat repetitive or overly detailed. A more concise summary that focuses on the clinical implications of the findings would improve readability and impact.

Minor Comments

• The English language has improved substantially, although a brief professional language check before publication would help ensure clarity and consistency throughout.

• Figures are clear and informative; however, a few could be slightly simplified for publication.

• If possible, please include a reference or note to the planned web-based calculator (e.g., “to be available at...”) before final submission.

Conclusion and Recommendation

The authors have satisfactorily addressed all major comments from the previous review. The methodological improvements—particularly the LASSO-based variable selection, inclusion of ACLF-specific comparators, and exclusion of patients with malignancies—are appropriate and have strengthened the validity of the model.

The remaining issues are minor and primarily of an editorial nature.

I therefore recommend acceptance after minor technical and language revisions.

**Do you want your identity to be public for this peer review?** For information about this choice, including consent withdrawal, please see our Privacy Policy

Reviewer #1: **Yes:** Lubomir Skladany

Reviewer #2: No

---

## [Author Response · Author response to Decision Letter 2]

7 Nov 2025

Response to Reviewers

Manuscript ID: PONE-D-25-36378R1

Title: Predicting Short-Term Mortality in Severe Cirrhosis: An Interpretable Machine Learning Model Integrating Routine Clinical Indicators

Dear Dr. Sona Frankova and Reviewers,

We are deeply grateful to you and the reviewers for the insightful feedback provided during the review of our manuscript. Your valuable comments have significantly enhanced the scientific rigor and overall quality of our work. In response, we have thoroughly considered all suggestions and have diligently addressed each point raised by the reviewers in our point-by-point response. For your convenience, all corresponding modifications in the manuscript have been clearly highlighted in the "Revised Manuscript with Track Changes" file.

Response to editor

Comment 1: Please include the following items when submitting your revised manuscript:

A rebuttal letter that responds to each point raised by the academic editor and reviewer(s). You should upload this letter as a separate file labeled 'Response to Reviewers'. A marked-up copy of your manuscript that highlights changes made to the original version. You should upload this as a separate file labeled 'Revised Manuscript with Track Changes'. An unmarked version of your revised paper without tracked changes. You should upload this as a separate file labeled 'Manuscript'.

Response 1: We appreciate the opportunity to submit a revised manuscript. We have carefully addressed all editor and reviewer feedback. In accordance with the guidelines, we are including our point-by-point response, a manuscript with highlighted changes, and an unmarked version.

Comment 2: Response 2: Thank you for your suggestions. Our Financial Disclosure Statement remains unchanged at this time. Besides, we have carefully addressed all points raised in the figure guidelines, we utilized NAAS, PLOS's free image analysis tool, to process our images in order to meet publication standards and each image has been revised accordingly. The specific changes are reflected in the updated figure files.

Comment 3: If applicable, we recommend that you deposit your laboratory protocols in protocols.io to enhance the reproducibility of your results. Protocols.io assigns your protocol its own identifier (DOI) so that it can be cited independently in the future. For instructions see: https://journals.plos.org/plosone/s/submission-guidelines#loc-laboratory-protocols. Additionally, PLOS ONE offers an option for publishing peer-reviewed Lab Protocol articles, which describe protocols hosted on protocols.io. Read more information on sharing protocols at https://plos.org/protocols?utm_medium=editorial-email&utm_source=authorletters&utm_campaign=protocols.

Response 3: We thank you for your advice. In accordance with your guidance, the laboratory protocol for this study has been deposited on protocols.io, where it has been assigned a citable Digital Object Identifier (DOI), provided below: DOI:10.17605/OSF.IO/FYNCG.

Comment 4: If the reviewer comments include a recommendation to cite specific previously published works, please review and evaluate these publications to determine whether they are relevant and should be cited. There is no requirement to cite these works unless the editor has indicated otherwise.

Response 4: Thank you for highlighting this important aspect of manuscript preparation. In our detailed point-by-point response to the reviewers, we have addressed all concerns raised. In this process, we noted that there were no specific directives from the reviewers concerning the addition of citations.

Comment 5: Please review your reference list to ensure that it is complete and correct. If you have cited papers that have been retracted, please include the rationale for doing so in the manuscript text, or remove these references and replace them with relevant current references. Any changes to the reference list should be mentioned in the rebuttal letter that accompanies your revised manuscript. If you need to cite a retracted article, indicate the article’s retracted status in the References list and also include a citation and full reference for the retraction notice.

Response 5: Thank you for your guidance. Following a thorough verification, we confirm that all references in our manuscript have been accurately sequenced and are correct. Furthermore, we have ensured that none of the cited articles have been retracted.

Responses to Reviewer #2:

Comment 1: External Validation. The model remains internally validated only. The authors now recognize this limitation and have described their plans for external validation, which is appreciated. However, the manuscript should emphasize even more clearly that the model is preliminary and not yet suitable for generalization beyond the MIMIC-IV ICU population. A short clarification in both the Abstract and Conclusions would be sufficient.

Response 1: Thank you very much for your valuable feedback. Our model has not yet undergone external validation, which will be an important part of our future research. Following your suggestion, we have clarified in both the Abstract and Conclusion sections that the model is currently not suitable for application outside the ICU population. You may refer to the Conclusion section of the Abstract in the manuscript, which states:

Conclusions: Our prediction model identifies high-risk cirrhotic patients and highlights critical prognostic factors, offering clinicians a valuable tool for risk stratification and timely intervention. The strong correlation between laboratory markers, complications, and outcomes underscores the importance of close monitoring in this population. However, our model is an initial step, effective within the ICU but requiring external, multi-center studies to broaden its clinical applicability, which is a clear priority for our future work.

Comment 2: Use of Extreme Laboratory Values. The justification for using maximum or minimum laboratory values has been provided, but this approach may introduce bias by incorporating information unavailable at admission. A brief note in the Limitations or Supplementary Material confirming that model performance was stable when admission values were tested would further support the robustness of the analysis.

Response 2: We sincerely thank you for this insightful comment. In response, we have performed an additional sensitivity analysis of our model. We recognized that employing maximum or minimum laboratory values in practical applications might introduce bias by incorporating information not available at the time of admission. To address this, we re-extracted the first available laboratory measurements after admission and incorporated these initial values into the model for validation. The results showed that the model using the initial laboratory values achieved an AUC of 0.771 in the training set and 0.767 in the validation set. Although these values are slightly lower than those of the model utilizing extreme values (training AUC = 0.837; validation AUC = 0.825), they remain well above 0.5 (Fig. 1). This indicates that the initial laboratory values still enable our model to predict short-term mortality effectively, thereby mitigating potential bias from missing admission-time data. Moreover, the model remains applicable for mortality prediction in patients with only a single laboratory measurement. We have elaborated on this aspect in the "Methodological Innovations" section of the manuscript as follows:

Methodological Innovations

This study is distinguished by three key advancements: the systematic integration of comorbidities, addressing oversights in previous models; the use of extreme-value laboratory analysis to capture critical physiological states more accurately than averaged measures; the range between maximum and minimum values may cause the mean to inaccurately reflect the patient's true clinical status. To mitigate this bias, we employed extreme analysis, which more reliably captures disease severity during hospitalization, although the use of extreme values may introduce bias by incorporating post-admission information. A post-hoc sensitivity analysis demonstrated stable predictive performance when only initial admission values were used (Training AUC: 0.771; Validation AUC: 0.767), thereby supporting the robustness of the model (S5 Fig); and the introduction of clinically actionable INR stratification, replacing traditional binary classifications. These innovations yielded a model demonstrating superior discrimination compared to existing scores while maintaining practicality through the use of routine parameters.

Fig 1. Sensitivity analysis of the prediction model. (A) ROC curves of the model in the training cohort. The blue curve represents the model using extreme laboratory values (AUC = 0.837), and the red curve represents the model using the first recorded values at admission (AUC = 0.771). (B) ROC curves of the model in the validation cohort. The blue curve corresponds to the model using extreme laboratory values (AUC = 0.825), and the red curve corresponds to the model using the first recorded values at admission (AUC = 0.767).

Comment 3: Scope and Discussion. The expanded discussion now provides a thoughtful pathophysiological explanation of bilirubin, INR, and hemolysis, which is appreciated. However, some sections remain somewhat repetitive or overly detailed. A more concise summary that focuses on the clinical implications of the findings would improve readability and impact.

Response 3: We sincerely thank you for this valuable suggestion. We agree that the previous description in this section was overly detailed and did not sufficiently highlight our key findings. Accordingly, we have thoroughly revised the relevant paragraph on page 26 of the manuscript. The updated text now reads:

Total bilirubin emerged as a significant predictor of short-term mortality in severe cirrhosis, a finding supported by its odds ratio (OR 1.027, 95% CI 1.002–1.052) in our model and consistent prognostic value in survival analysis (HR 1.023, 95% CI 1.010–1.037, p = 0.001). The elevation in total bilirubin reflects both hepatic and extrahepatic pathophysiological processes. On one hand, portal hypertension-induced hypersplenism promotes chronic hemolysis through splenic sequestration and destruction of red blood cells, exacerbated by altered hemorheological properties and microvascular stress. This leads to increased indirect bilirubin from hemoglobin breakdown. On the other hand, intrahepatic cholestasis and impaired hepatocyte function reduce bilirubin clearance, elevating direct bilirubin levels. Additionally, in decompensated cirrhosis, diminished glucuronidation capacity further contributes to the accumulation of indirect bilirubin. Together, these mechanisms underpin the strong association between elevated total bilirubin and mortality, highlighting its utility in risk stratification and early intervention for high-risk cirrhotic patients.

Response to Minor Comments

Comment 1: The English language has improved substantially, although a brief professional language check before publication would help ensure clarity and consistency throughout.

Response 1: Thank you for this suggestion. We have thoroughly reviewed the language throughout the manuscript and have revised several statements to ensure greater scientific clarity and precision.

Comment 2: Figures are clear and informative; however, a few could be slightly simplified for publication.

Response 2: We thank the reviewer for this suggestion. We agree that the original figures had some limitations. Accordingly, all figures have been revised to ensure they meet the standards for publication in terms of clarity and presentation.

Comment 3: If possible, please include a reference or note to the planned web-based calculator (e.g., "to be available at...") before final submission.

Response 3: We sincerely thank you for this suggestion. In response, we have developed a preliminary web-based calculator for our prediction model, which is now accessible at: https://9zhangshun.shinyapps.io/livercirrhosis/. Additionally, we have added a corresponding statement in the "Future Directions" section of the manuscript to inform readers of this resource.

---

## [Editor Report · Decision Letter 2]

18 Nov 2025

Predicting Short-Term Mortality in Severe Cirrhosis: An Interpretable Machine Learning Model Integrating Routine Clinical Indicators

PONE-D-25-36378R2

Dear Dr. Wen,

We’re pleased to inform you that your manuscript has been judged scientifically suitable for publication and will be formally accepted for publication once it meets all outstanding technical requirements.

Kind regards,

Sona Frankova

Academic Editor

PLOS ONE
---

## [Editor Report · Acceptance letter]

PONE-D-25-36378R2

PLOS One

Dear Dr. Wen,

I'm pleased to inform you that your manuscript has been deemed suitable for publication in PLOS One. Congratulations! Your manuscript is now being handed over to our production team.

Kind regards,

on behalf of

Dr. Sona Frankova

Academic Editor

PLOS One